# Tensor Product Neural Networks for Functional ANOVA Model

**Seokhun Park** [1]  **Insung Kong** [2]  **Yongchan Choi** [3]  **Chanmoo Park** [1 4]  **Yongdai Kim** [1]

## Abstract

Interpretability for machine learning models is becoming more and more important as machine learning models become more complex. The functional ANOVA model, which decomposes a high-dimensional function into a sum of lower dimensional functions (commonly referred to as *components*), is one of the most popular tools for interpretable AI, and recently, various neural networks have been developed for estimating each component in the functional ANOVA model. However, such neural networks are highly unstable when estimating each component since the components themselves are not uniquely defined. That is, there are multiple functional ANOVA decompositions for a given function. In this paper, we propose a novel neural network which guarantees a unique functional ANOVA decomposition and thus is able to estimate each component stably. We call our proposed neural network *ANOVA Tensor Product Neural Network (ANOVA-TPNN)* since it is motivated by the tensor product basis expansion. Theoretically, we prove that ANOVA-TPNN can approximate any smooth function well. Empirically, we show that ANOVA-TPNN provide much more stable estimation of each component and thus much more stable interpretation when training data and initial values of the model parameters vary than existing neural networks do. Our source code is released at https://github.com/ParkSeokhun/ANOVA-TPNN

## 1. Introduction

Interpretability has become more important as artificial intelligence (AI) models have become more sophisticated and complicated in recent years. Various methods such as LIME

(Ribeiro et al., 2016) and SHAP (Lundberg, 2017) have been suggested to interpret complex black-box AI models. However, these methods explain a given black-box model through a locally approximated interpretable models and thus often fail to provide a faithful global view of the model (Slack et al., 2020).

The functional ANOVA model, which approximates a given complex high-dimensional function by the sum of low-dimensional (e.g., one or two dimensional) functions, is a well known transparent-box AI model. One of the most representative examples of the functional ANOVA model is the generalized additive model (GAM, Hastie & Tibshirani (1987)), which consists of the sum of one-dimensional functions, each corresponding to each input feature. Low-dimensional functions are easier to understand, and thus the functional ANOVA model is popularly used for interpretable AI (Lengerich et al., 2020; Märtens & Yau, 2020).

Recently, several specially designed neural networks for the functional ANOVA model have been proposed, including NAM (Agarwal et al., 2021) and NBM (Radenovic et al., 2022). These neural networks can be learned by standard stochastic gradient descent algorithms and thus can be applied to large sized data compared to traditional learning algorithms based on basis expansions (Wood, 2006) and regularizations (Gu & Wahba, 1993). However, existing neural networks are not good at estimating each component (each low-dimensional function in the functional ANOVA model) mainly due to unidentifiability. Here, 'unidentifiability' means that there exist multiple different functional ANOVA decompositions of a given function and we do not know which decomposition a gradient descent algorithm converges on. Note that poor estimation of the components would result in poor interpretation.

In this paper, we develop new neural networks for the functional ANOVA model such that each component is identifiable but they are learnable by standard stochastic gradient descent algorithms. Identifiability makes our proposed neural networks be good at estimating the components and thus provide reliable interpretation. In addition, it is robust to outliers and easy to reflect monotone constraints.

To develop the proposed neural networks, we begin with the tensor product basis expansion (Wood, 2006) and replace each basis function by specially designed neural networks so

---

[1]Department of Statistics, Seoul National University, Seoul, Republic of Korea [2]Department of Applied Mathematics, University of Twente, Enschede, Netherlands [3]Toss Bank, Seoul, Republic of Korea [4]Samsung Electronics Co., Ltd, Republic of Korea. Correspondence to: Yongdai Kim <ydkim0903@gmail.com>.

*Proceedings of the 42nd International Conference on Machine Learning*, Vancouver, Canada. PMLR 267, 2025. Copyright 2025 by the author(s).

that each component becomes identifiable and robust to outliers. We call our proposed neural networks *Tensor Product Neural Networks* (TPNN). Finally, we propose ANOVA-TPNN, which estimates each component in the functional ANOVA model using TPNNs. Theoretically, we prove that ANOVA-TPNN has the universal approximation property in the sense that it approximates any Lipschitz function well.

Overall, our contributions are summarized as follows.

- We propose novel neural networks (TPNN) for the functional ANOVA model with which we can estimate each component stably and accurately by use of a standard stochastic gradient descent algorithm.

- We prove the universal approximation property in the sense that TPNN can approximate any Lipschitz function up to an arbitrary precision.

- By analyzing multiple benchmark datasets, we demonstrate that TPNN provides more accurate and stable estimation and interpretation of each component compared to the baseline models, including NAM (Agarwal et al., 2021), NBM (Radenovic et al., 2022), NODE-GAM (Chang et al., 2021) and XGB (Chen & Guestrin, 2016) without losing prediction accuracy.

## 2. Background

### 2.1. Notation

Let $\mathbf{x} = (x_1, ..., x_p)^\top \in \mathcal{X} = \mathcal{X}_1 \times ... \times \mathcal{X}_p$ be a vector of input features, where we assume $\mathcal{X} \subseteq [-a, a]^p$ for some $a > 0$. We denote $[p] = \{1, \ldots, p\}$, and denote its power set as $\mathcal{P}([p])$. For $\mathbf{x} \in \mathcal{X}$ and $S \subseteq [p]$, let $\mathbf{x}_S = (x_j, j \in S)^\top$. We denote $f_S$ as a function of $\mathbf{x}_S$. For a real-valued function $f : \mathcal{X} \to \mathbb{R}$, we denote $||f||_\infty = \sup_{\mathbf{x} \in \mathcal{X}} |f(\mathbf{x})|$.

### 2.2. Functional ANOVA model

The functional ANOVA model (Hoeffding, 1992; Sobol, 2001) decomposes a high-dimensional function $f$ into the sum of low-dimensional functions

$$f(\mathbf{x}) = \beta_0 + \sum_{j=1}^{p} f_j(x_j) + \sum_{j<k} f_{jk}(x_j, x_k) + \cdots,$$

which is considered as one of the most important XAI tools Agarwal et al. (2021); Radenovic et al. (2022); Lengerich et al. (2020). Typically, $f$ is defined as $f(\mathbf{x}) = g(\mathbb{E}(Y|X = x))$ where $g$ is a link function and $Y$ is target variable. In the functional ANOVA model, $f_j, j \in [p]$ are called the main effects, and $f_{j,k}, (j, k) \in [p]^2$ are called the second interaction terms and so on. In practice, only interactions of lower orders (e.g., the main and second order only) are included in the decomposition for easy interpretation.

The Generalized Additive Model (GAM, Hastie & Tibshirani (1987)) is a special case of the functional ANOVA

model with only main effects included, that is

$$f(\mathbf{x}) = \beta_0 + \sum_{j=1}^{p} f_j(x_j).$$

Similarly, $GA^2M$ is defined as the functional ANOVA model including all of the main effects and second order interactions,

$$f(\mathbf{x}) = \beta_0 + \sum_{S \subseteq [p], |S| \leq 2} f_S(\mathbf{x}_S),$$

or more generally we can consider $GA^dM$ defined as

$$f(\mathbf{x}) = \beta_0 + \sum_{S \subseteq [p], |S| \leq d} f_S(\mathbf{x}_S).$$

Several learning algorithms for the functional ANOVA model have been proposed. Gu & Wahba (1993) applied the smoothing spline to learn the functional ANOVA model, Lin & Zhang (2006) developed a component-wise sparse penalty, Kim et al. (2009) proposed a boosting algorithm for the functional ANOVA model, and Wood (2006); Rügamer (2024) proposed methods for estimating the functional ANOVA model using basis expansion.

In addition, the functional ANOVA model has been applied to various problems such as sensitivity analysis (Chastaing et al., 2012), survival analysis (Huang et al., 2000), diagnostics of high-dimensional functions (Hooker, 2007) and machine learning models (Lengerich et al., 2020; Märtens & Yau, 2020).

Recently, learning the functional ANOVA model using neural networks has received much attention since gradient descent based learning algorithms can be easily scaled up. Examples are Neural Additive Model (NAM, Agarwal et al. (2021)), Neural Basis Model (NBM, Radenovic et al. (2022)) and NODE-GAM (Chang et al., 2021) to name just a few. NAM models each component of GAM by DNNs, and NBM achieves a significant reduction in training time compared to NAM by using basis DNNs to learn all components simultaneously. NODE-GAM extends Neural Oblivious Decision Ensembles (NODE, Popov et al. (2019)) for GAM.

### 2.3. Tensor Product Basis

Let $\mathcal{B}_j = \{B_{j,k}(\cdot), k = 1, \ldots, M_j\}$ be a given set of basis functions on $\mathcal{X}_j$ (e.g. truncated polynomials or B-splines). Then, the main effect is approximated as follows:

$$f_j(x_j) \approx \sum_{k=1}^{M_j} \alpha_{jk} B_{j,k}(x_j)$$
$$= \mathbf{B}_j(x_j)^\top \boldsymbol{\alpha}_j.$$

where $\mathbf{B}_j(x_j) = (B_{j,1}(x_j), ..., B_{j,M_j}(x_j))^\top$ and $\boldsymbol{\alpha}_j = (\alpha_{j,1}, ..., \alpha_{j,M_j})^\top$.

For the second order interactions, $f_{jk}(x_j, x_k)$ is approximated by

$$
\begin{aligned}
f_{jk}(x_j, x_k) &\approx (\mathbf{B}_j(x_j) \otimes \mathbf{B}_k(x_k))\boldsymbol{\alpha}_{j,k} \\
&= \sum_{l=1}^{M_j} \sum_{h=1}^{M_k} B_{j,l}(x_j) B_{k,h}(x_k) \alpha_{j,k,l,h}.
\end{aligned}
$$

Here, $\mathbf{B}_j(x_j) \otimes \mathbf{B}_k(x_k) = (B_{j,l}(x_j)B_{kh}(x_k))_{l \in [M_l], k \in [M_k]}$ are called the tensor product basis functions and $\boldsymbol{\alpha}_{j,k} = (\alpha_{j,k,l,h}, l \in [M_l], k \in [M_k])^\top \in \mathbb{R}^{M_l M_k}$.

Similarly, for $S \subseteq [p]$, a general order interaction $f_S(\mathbf{x}_S)$ is approximated by

$$
f_S(\mathbf{x}_S) \approx (\otimes_{j \in S} \mathbf{B}_j(x_j))\boldsymbol{\alpha}_S,
$$

where $\otimes_{j \in S} \mathbf{B}_j(x_j)$ and $\boldsymbol{\alpha}_S$ are $\prod_{j \in S} M_j$-dimensional tensors.

A problem of using tensor product basis functions is that the number of learnable parameters (i.e. $\alpha_{Si_1 \cdots i_{|S|}}, i_j \in [M_j]$) increases exponentially in $|S|$ and thus estimating higher order interactions would be computationally demanding. Thus, they are typically used for low-dimensional datasets.

# 3. Tensor Product Neural Networks

The aim of this section is to propose a specially designed neural models so called Tensor Product Neural Networks (TPNN). The most important advantage of TPNN is that they satisfy an identifiability condition of each component in the functional ANOVA model and so we can estimate each component accurately and stably by use of a gradient descent algorithm. Additionally, unlike the conventional tensor product basis approach (Wood, 2006; Rügamer, 2024), our model does not lead to an exponential increase in the number of learnable parameters with respect to $|S|$, making it applicable to high-dimensional datasets. Finally, TPNN is robust to input outliers and easy to accommodate the monotone constraint on the main effects.

In this section, we first explain the sum-to-zero condition for the identifiability of each component in the functional ANOVA model and then propose TPNNs that always satisfy the sum-to-zero condition, and develop a corresponding learning algorithm for the functional ANOVA model. Moreover, we show theoretically that TPNNs are flexible enough so that they can approximate any Lipschitz continuous function well. A modification of Neural Basis Models with TPNNs is also discussed.

## 3.1. Sum-to-zero condition for identifiable components

The functional ANOVA model itself is not identifiable. That is, there are multiple functional ANOVA decompositions for a given function. For example,

$$
f(x_1, x_2) = f_1(x_1) + f_2(x_2) + f_{12}(x_1, x_2)
$$

where $f_1(x_1) = x_1, f_2(x_2) = x_2, f_{1,2}(x_1, x_2) = x_1 x_2$ can be expressed as

$$
f(x_1, x_2) = f_1^*(x_1) + f_2^*(x_2) + f_{1,2}^*(x_1, x_2)
$$

where $f_1^*(x_1) = -x_1, f_2^*(x_2) = x_2, f_{12}^*(x_1, x_2) = x_1(x_2 + 2)$ (Lengerich et al., 2020). Without the identifiability condition, each component cannot be estimated uniquely and thus interpretation through the estimated components becomes unstable. See empirical evidences of instability of estimated components by NAM Agarwal et al. (2021) and NBM Radenovic et al. (2022) are given in Appendix E.1.

A simple remedy to ensure the identifiability of each component is to put a constraint. One of the most popular constraints for identifiability of each component in the functional ANOVA model is so called the *sum-to-zero* condition (Hooker, 2007; Kim et al., 2009). The *sum-to-zero* condition requires that each component $f_S$ should satisfy

$$
\begin{aligned}
&\forall j \in S, \ \forall \mathbf{z} \in \mathcal{X}_{S \setminus \{j\}}, \\
&\int_{\mathcal{X}_j} f_S(\mathbf{x}_{S \setminus \{j\}} = \mathbf{z}, x_j)\mu_j(dx_j) = 0
\end{aligned} \tag{1}
$$

for some probability measure $\mu_j$ on $\mathcal{X}_j$. For $\mu_j$, the empirical distribution of the input feature $x_j$ or the uniform distribution on $\mathcal{X}_j$ can be used in practice. With the sum-to-zero in (1), the functional ANOVA model becomes identifiable, as can be seen in proposition 3.1. Let $\mu = \prod_j \mu_j$.

**Proposition 3.1.** *(Hooker, 2007) Consider two component sets $\{f_S^1, S \subseteq [p], |S| \leq d\}$ and $\{f_S^2, S \subseteq [p], |S| \leq d\}$ which satisfy (1). Then, $\sum_{S:|S| \leq d} f_S^1(\cdot) \equiv \sum_{S:|S| \leq d} f_S^2(\cdot)$ almost everywhere (with respect to $\mu$) if and only if $f_S^1(\cdot) \equiv f_S^2(\cdot)$ almost everywhere (with respect to $\mu$) for every $S \subseteq [p]$ with $|S| \leq d$.*

Herren & Hahn (2022) demonstrated that there is an interesting relation between the sum-to-zero condition and SHAP (Lundberg, 2017) which is a well known interpretable AI method. That is, the SHAP value of a given input can be calculated easily from the prediction values of each component of the functional ANOVA model. For a given function $f$ and input vector $\mathbf{x}$, the SHAP value of the $j$th input variable is defined as

$$
\phi_j(f, \mathbf{x}) = \sum_{S \subseteq [p] \setminus \{j\}} \frac{|S|!(p - |S| - 1)!}{p!} (v_f(\mathbf{x}_{S \cup \{j\}}) - v_f(\mathbf{x}_S)),
$$

where $v_f(\mathbf{x}_S) = \mathbb{E}[f(\mathbf{X})|\mathbf{X}_S = \mathbf{x}_S]$, where $\mathbf{X} \sim \mu$.

**Proposition 3.2.** *(Herren & Hahn, 2022) For a given $GA^dM$ $f$ satisfying the sum-to-zero condition, we have*

$$
\phi_j(f, \mathbf{x}) = \sum_{S \subseteq [p], |S| \leq d, j \in S} f_S(\mathbf{x}_S)/|S|. \tag{2}
$$

Equation (2) provides an interesting implication - the functional ANOVA model satisfying the sum-to-zero condition also decomposes the SHAP value. That is, the contribution of the interaction between $x_j$ and $\mathbf{x}_S$ to the SHAP

value $\phi_j(f, \mathbf{x})$ is $f(\mathbf{x}_{S'})/|S'|$, where $S' = S \cup \{j\}$. This result could provide a new way of calculating the SHAP value approximately. Specifically, we first approximate a given function by a GA$^d$M satisfying sum-to-zero condition and calculate the SHAP value of the approximated GA$^d$M by use of (2). We refer this approximated SHAP value as *ANOVA-SHAP*. The results of numerical studies in Appendix H confirm that ANOVA-SHAP is similar to Kernel-SHAP calculated from the shap python package (Lundberg, 2017) One obvious advantage of ANOVA-SHAP is that it is significantly faster to compute compared to existing methods of calculating SHAP values such as Deep-SHAP and Kernel-SHAP proposed by Lundberg (2017), as well as Tree-SHAP proposed by Lundberg et al. (2018).

### 3.2. ANOVA-TPNN

We first propose basis neural networks for the main effects and extend these basis neural networks for higher order interactions through tensor product which we call tensor product neural networks (TPNN). Finally, we propose ANOVA-TPNN which estimates each component by TPNN.

**Main effects.** We first consider the main effects. Let $\phi_j(x|\theta)$ be a basis neural networks for the main effect $f_j$ parametrized by $\theta$. That is, we approximate $f_j(x_j)$ by

$$f_j(x_j) \approx \sum_{k=1}^{K_j} \beta_{jk} \phi_j(x_j|\theta_{j,k}) \tag{3}$$

for $\beta_{jk} \in \mathbb{R}$. To ensure the sum-to-zero condition, we choose $\phi_j$ such that $\int_{x \in \mathcal{X}_j} \phi_j(x|\theta)\mu_j(dx) = 0$ for all $\theta$.

Our choice of $\phi_j(x|\theta)$ is as follows:

$$\phi_j(x|\theta) = \left\{ 1 - \sigma\left(\frac{x-b}{\gamma}\right) \right\} + c_j(b, \gamma)\sigma\left(\frac{x-b}{\gamma}\right)$$

for $b \in \mathbb{R}$ and $\gamma > 0$, where $\theta = (b, \gamma)$, $\sigma(\cdot)$ is an activation function and $c_j(b, \gamma) = -(1 - \eta_j(b, \gamma))/\eta_j(b, \gamma)$ with $\eta_j(b, \gamma) = \int_{x \in \mathcal{X}_j} \sigma\left(\frac{x-b}{\gamma}\right) \mu_j(dx)$. We introduce the term $c_j(b, \gamma)$ to ensure $\phi_j(x|\theta)$ satisfies the sum-to-zero condition. For $\sigma(\cdot)$ function, any differentiable function can be used but in this paper we use the sigmoid function for $\sigma(x)$ (i.e. $\sigma(x) = 1/(1 + \exp(-x))$), since it is robust to input outliers and provides a nice theoretical property: an universal approximation theorem given in Section 3.3.

The standard basis function approach can be understood as a method of fixing the parameters $\theta_{jk}$s a priori and only learning $\beta_{jk}$s in (3). In contrast, we learn $\theta_{jk}$ as well as $\beta_{jk}$. That is, the terms $\phi_j(x|\theta_{j,k})$ can be considered as data-adaptive basis functions. Since there is no constraint (except the nonnegative constraint on $\gamma$) on the parameters, a gradient descent algorithm can be used for learning. The number of basis functions $K_j$ is fixed in advance as is done in the basis expansion approaches.

**Higher order interactions.** For $f_S$, we consider the following tensor product neural network (TPNN)

$$f_S(\mathbf{x}_S) \approx \sum_{k=1}^{K_S} \beta_{S,k} \phi_S(\mathbf{x}_S|\theta_{S,k}) \tag{4}$$

where $\phi_S(\mathbf{x}_S|\theta_{S,k}) = \prod_{j \in S} \phi_j(x_j|\theta_{j,k}^S)$. Since $\phi_j(x_j|\theta_{j,k}^S)$ satisfies the sum-to-zero condition, $\phi_S(\mathbf{x}_S|\theta_{S,k})$ does and so does $f_S$. As is done for the main effect, we learn $\beta_{S,k}$s and $\theta_{S,k}$s while we let $K_S$ a hyper-parameter.

**Remark.** As the traditional tensor product basis expansion approaches (Wood, 2006; Rügamer, 2024) do, we may consider

$$f_S(\mathbf{x}_S) \approx (\otimes_{j \in S} \Phi_j(x_j))\boldsymbol{\beta}_S,$$

where $\Phi_j(x_j) = (\phi_j(x_j|\theta_{j,k}), k \in [K_j])$, $\boldsymbol{\beta}_S$ is a $\prod_{j \in S} K_j$-dimensional tensor, and $\theta_{j,k}$s are those used for the main effect. This expansion shares the parameters between the components and thus the number of parameters $\theta_{jk}$s is smaller than the expansion in (4). However, the number of parameters in $\boldsymbol{\beta}_S$ is exponentially proportional to $|S|$ and thus the number of the total learnable parameters would be much larger than that in the expansion (4). This is an important advantage of the proposed TPNNs compared to the traditional tensor product basis expansions.

**ANOVA-T$^d$PNN.** Finally, we propose ANOVA-T$^d$PNN that estimates each component of the functional ANOVA model using TPNN as follows.

$$f(\mathbf{x}) = \beta_0 + \sum_{S:|S| \leq d} \sum_{k=1}^{K_S} \beta_{S,k} \phi_S(\mathbf{x}_S|\theta_{S,k}).$$

In ANOVA-T$^d$PNN, the learnable parameters are $\beta_0$ and $(\beta_{S,k}, \theta_{S,k}), k \in [K_S], S \subseteq [p], |S| \leq d$. Unless there is any confusion, we use ANOVA-TPNN and ANOVA-T$^d$PNN for general $d$ interchangeably.

**Training.** For given training data $(\mathbf{x}_1, y_1), \dots, (\mathbf{x}_n, y_n)$ and a given loss function $\ell$, we learn the parameters by minimizing the empirical risk $\sum_{i=1}^n \ell(y_i, f(\mathbf{x}_i))$ by a gradient descent algorithm. Overfitting can be avoided by selecting the number of epochs and learning rate carefully.

**Data preprocessing.** The term $c_j(\alpha, \gamma)$ could be too large when $\eta(b, \gamma)$ is close to 0, which can happen when $\mu_j$ is the empirical distribution and there exist outliers. To avoid this problem, we transform each input feature based on the marginal ranks to make $\mu_j$ of the transformed data be similar to the uniform distribution. Since ANOVA-TPNN is nonparametric, this data transformation does not affect much to the final prediction model.

Table 1: **Stability scores on real datasets.** Lower stability score means more stable interpretation. The bold faces highlight the best results among GAM and GA²M.

| Dataset | GA¹M | | | GA²M | | |
|---|---|---|---|---|---|---|
| | ANOVA T¹PNN | NA¹M | NB¹M | ANOVA T²PNN | NA²M | NB²M |
| CALHOUSING (Pedregosa et al., 2011a) | **0.012** | 0.045 | 0.039 | **0.035** | 0.071 | 0.075 |
| WINE(Cortez et al., 2009) | **0.011** | 0.058 | 0.043 | **0.049** | 0.087 | 0.065 |
| ONLINE (Fernandes et al., 2015) | **0.031** | 0.076 | 0.054 | **0.052** | 0.072 | 0.072 |
| ABALONE (Warwick et al., 1995) | **0.008** | 0.013 | 0.026 | **0.028** | 0.047 | 0.038 |
| FICO (fic, 2018) | **0.035** | 0.046 | 0.046 | **0.048** | 0.089 | 0.075 |
| CHURN(chu, 2017) | **0.017** | 0.027 | 0.047 | **0.047** | 0.089 | 0.080 |
| CREDIT (cre, 2015) | **0.021** | 0.069 | 0.025 | **0.036** | 0.089 | 0.053 |
| LETTER (Slate, 1991) | 0.017 | 0.022 | **0.014** | **0.026** | 0.075 | 0.081 |
| DRYBEAN(dry, 2020) | **0.028** | 0.074 | 0.070 | **0.053** | 0.088 | 0.081 |

### 3.3. Universal approximation thorem

An interesting theoretical property of ANOVA-TPNN is the universal approximation property as the standard neural network has (Hornik et al., 1989). That is, ANOVA-TPNN can approximate any arbitrary GA$^d$M function to a desired level of accuracy, as stated in the following theorems.

**Theorem 3.3.** *Suppose that $\mu_j, j \in [p]$ have lower and upper bounded densities with respect to the Lebesgue measure. Then, for any L-Lipschitz continuous function[1] $g_{0,S} : \prod_{j \in S} \mathcal{X}_j \to \mathbb{R}$ satisfying the sum-to-zero condition, there exists a TPNN with $K_S$ many basis neural networks such that*

$$\left\| g_{0,S}(\cdot) - \sum_{k=1}^{K_S} \beta_{S,k} \phi_S(\cdot | \theta_{S,k}) \right\|_\infty < C_S \frac{|S|}{K_S^{\frac{1}{|S|}} + 1}$$

*for some constant $C_S > 0$ and $S \subseteq [p]$.*

Theorem 3.3 shows that TPNN can approximate any Lipschitz continuous function satisfying the sum-to-zero condition. with an arbitrary precision by choosing $K_S$ sufficiently large, and the required $K_S$ for a given precision should increase as $|S|$ increases. An obvious corollary of Theorem 3.3 is that ANOVA-TPNN can approximate any GA$^d$M model where each component is Lipschitz continuous and satisfies the sum-to-zero condition.

**Corollary 3.4.** *Let $g_0(\mathbf{x}) := \sum_{S \subseteq [p], |S| \le d} g_{0,S}(\mathbf{x}_S)$ be a given GA$^d$M function satisfying the sum-to-zero condition. If $\mu_j, j \in [p]$ be probability measures having bounded densities with respect to the Lebesgue measure. and each $g_{0,S}$ is L-Lipschitz continuous, then, there exists $f_{ANOVA-T^dPNN}$ such that*

$$\left\| g_0(\cdot) - f_{ANOVA-T^dPNN}(\cdot) \right\|_\infty < C \sum_{S \subseteq [p], |S| \le d} \frac{|S|}{K_S^{\frac{1}{|S|}} + 1}$$

*for some constant $C > 0$, where $K_S$ is the number of basis neural networks for component $S$.*

---

[1] A given function $v$ defined on $\mathcal{Z}$ is $L$-Lipschitz continuous if $|v(z_1) - v(z_2)| \le L\|z_1 - z_2\|$ for all $z_1, z_2 \in \mathcal{Z}$, where $\| \cdot \|$ is a certain norm defined on $\mathcal{Z}$.

### 3.4. Extension to Neural Basis Models

Similarly to NBM (Radenovic et al., 2022), we extend ANOVA-TPNN to NBM-TPNN, which estimates each component as a linear combination of common basis neural networks.

Let $\mathcal{X}_j, j \in [p]$ be all equal to $\mathcal{X}_0$ (e.g. $\mathcal{X}_0 = [-1, 1]$) and let $\mu_j, j \in [p]$ be also all equal to $\mu_0$ (e.g. the uniform distribution on $[-1, 1]$). Let $\phi(x|\theta)$ be a basis neural network on $\mathcal{X}_0$ satisfying the sum-to-zero condition with respect to $\mu_0$. Then, NBM-TPNN approximates $f_S$ by

$$f_S(\mathbf{x}_S) \approx \sum_{k=1}^{K} \beta_{S,k} \prod_{j \in S} \phi(x_j | \theta_{|S|,k}).$$

That is, NBM-TPNN shares basis neural networks across the components with the same cardinality, which substantially reduces the number of learnable parameters. The experimental results for NBM-TPNN are provided in Section 4.6.

## 4. Experiments

This section presents the results of numerical experiments. More results along with details about data, algorithms and selection of hyper-parameters are provided in Appendices B to M.

### 4.1. Stability in component estimation

Similarly to NAM (Agarwal et al., 2021) and NBM (Radenovic et al., 2022), ANOVA-TPNN provides interpretation through the estimated components. Thus, if the components are not estimated stably, the interpretations based on the estimated components would not be reliable. In this subsection, we investigate the stability of the component estimation of ANOVA-TPNN compared with the other baseline models including NAM and NBM. For this purpose, we generate randomly sampled training data and estimate the components of the functional ANOVA model. We repeat this procedure 10 times to obtain 10 estimates of each component, and measure how similar these 10 estimates are. For the similarity measure, we use

Table 2: **Performance of component selection.** We report the averages (standard deviations) of AUROCs of the estimated importance scores of each component on $f^{(1)}, f^{(2)}, f^{(3)}$ synthetic datasets. The bold faces highlight the best results.

| True model | $f^{(1)}$ | | | $f^{(2)}$ | | | $f^{(3)}$ | | |
|---|---|---|---|---|---|---|---|---|---|
| Models | ANOVA T$^2$PNN | NA$^2$M | NB$^2$M | ANOVA T$^2$PNN | NA$^2$M | NB$^2$M | ANOVA T$^2$PNN | NA$^2$M | NB$^2$M |
| AUROC ↑ | **1.000** (0.00) | 0.330 (0.08) | 0.522 (0.16) | **0.943** (0.01) | 0.311 (0.08) | 0.481 (0.09) | **0.956** (0.02) | 0.381 (0.13) | 0.477 (0.07) |

Table 3: **Prediction performance.** We report the averages (standard deviations) of the prediction performance measure. In addition, we report the averages of ranks of prediction performance of each model on nine datasets. The optimal (or suboptimal) results are highlighted in **bold** (or underlined).

| Dataset | Measure | GA$^1$M | | | | GA$^2$M | | | | Black box | |
|---|---|---|---|---|---|---|---|---|---|---|---|
| | | ANOVA T$^1$PNN | NODE GA$^1$M | NA$^1$M | NB$^1$M | ANOVA T$^2$PNN | NODE GA$^2$M | NA$^2$M | NB$^2$M | XGB | DNN |
| CALHOUSING | RMSE ↓ | 0.614 (0.01) | 0.581 (0.01) | 0.659 (0.01) | 0.594 (0.08) | 0.512 (0.01) | 0.515 (0.01) | 0.525 (0.02) | 0.502 (0.03) | **0.452** (0.01) | 0.518 (0.01) |
| WINE | RMSE ↓ | 0.725 (0.02) | 0.723 (0.02) | 0.733 (0.02) | 0.724 (0.02) | 0.704 (0.02) | 0.730 (0.02) | 0.720 (0.02) | 0.702 (0.03) | **0.635** (0.03) | 0.696 (0.01) |
| ONLINE | RMSE ↓ | **1.111** (0.25) | 1.121 (0.27) | 1.350 (0.57) | 1.187 (0.25) | **1.111** (0.25) | 1.137 (0.26) | 1.313 (0.46) | 1.179 (0.21) | 1.122 (0.26) | 1.123 (0.26) |
| ABALONE | RMSE ↓ | 2.135 (0.09) | 2.141 (0.09) | 2.171 (0.08) | 2.167 (0.09) | 2.087 (0.08) | 2.100 (0.10) | 2.088 (0.08) | 2.088 (0.08) | 2.164 (0.09) | **2.071** (0.10) |
| FICO | AUROC ↑ | 0.799 (0.007) | 0.795 (0.009) | 0.788 (0.006) | 0.797 (0.006) | **0.800** (0.007) | 0.793 (0.007) | 0.799 (0.007) | 0.799 (0.008) | 0.796 (0.008) | 0.793 (0.008) |
| CHURN | AUROC ↑ | 0.839 (0.012) | 0.824 (0.012) | **0.846** (0.011) | 0.845 (0.012) | 0.842 (0.012) | 0.830 (0.011) | 0.844 (0.011) | 0.844 (0.011) | **0.846** (0.012) | 0.842 (0.013) |
| CREDIT | AUROC ↑ | 0.983 (0.005) | 0.983 (0.005) | 0.976 (0.012) | 0.972 (0.011) | 0.984 (0.006) | **0.985** (0.006) | 0.980 (0.007) | **0.985** (0.004) | 0.983 (0.004) | 0.980 (0.006) |
| LETTER | AUROC ↑ | 0.900 (0.003) | 0.910 (0.002) | 0.904 (0.001) | 0.910 (0.001) | 0.984 (0.001) | 0.988 (0.001) | 0.986 (0.001) | 0.990 (0.001) | **0.997** (0.001) | 0.996 (0.001) |
| DRYBEAN | AUROC ↑ | 0.995 (0.001) | 0.996 (0.001) | 0.996 (0.001) | 0.994 (0.001) | **0.998** (0.001) | 0.996 (0.001) | 0.995 (0.001) | 0.995 (0.001) | 0.997 (0.001) | 0.997 (0.001) |
| | Rank avg ↓ | 6.22 | 5.44 | 8.11 | 7.44 | **3.11** | 5.56 | 5.33 | 3.56 | **3.11** | 4.33 |

$$\mathcal{SC}(f_S) = \frac{1}{n} \sum_{i=1}^{n} \frac{\sum_{j=1}^{10} (f_S^j(\mathbf{x}_i) - \bar{f}_S(\mathbf{x}_i))^2}{\sum_{j=1}^{10} (f_S^j(\mathbf{x}_i))^2}$$

for given pre-selected $n$ many input vectors $\mathbf{x}_i$, $i = 1, \ldots, n$, where $f_S^j$, $j = 1, \ldots, 10$ are the 10 estimates of $f_S$ and $\bar{f}_S$ is their average. A smaller value of $\mathcal{SC}(f_S)$ means a more stable estimation (and thus more reliable interpretation).

We compare the overall stability score $\mathcal{SC}(f) = \sum_S \mathcal{SC}(f_S)/|S|$. For each of nine benchmark datasets, we calculate the overall stability scores of ANOVA-TPNN, NAM, and NBM, whose results are given in Table 1. The results again confirm that ANOVA-TPNN is superior in terms of the stability of component estimation.

Also, the plots of the functional relations of the main effects are provided in Appendix E.1, which amply demonstrate the instability of NAM and NBM in component estimation.

The stability of ANOVA-TPNN with respect to the choice of initial values are illustrated in Appendix C.2.

### 4.2. Performance in component selection

To investigate how well ANOVA-TPNN selects the true signal components, we conduct an experiment similar to the one in Tsang et al. (2017). We consider the $l_1$ norm of each estimated component (i.e, $\|f_S(\mathbf{x}_S)\|_1$) as the important score, and select the components whose important scores are large. We generate synthetic datasets from $Y = f^k(\mathbf{x}) + \epsilon$, where $f^k$ is the true prediction model defined in Appendix B.1 for $k = 1, 2, 3$. Then, we apply ANOVA-T$^2$PNN, NA$^2$M and NB$^2$M to calculate the importance scores of the main effects and second order interactions and examine how well they predict whether a given component is signal. Table 2 compares the AUROCs of ANOVA-T$^2$PNN, NA$^2$M and NB$^2$M, which clearly indicates that ANOVA-T$^2$PNN outperforms the baseline models in component selection. More details regarding component selection with ANOVA-T$^2$PNN are given in Appendix B.4.

### 4.3. Prediction performance

We compare prediction performance of ANOVA-TPNN with baseline models. We randomly split the train, validation and test data into the ratio 70/10/20, where the validation data is used to select the optimal epoch and the test data is used to measure the prediction performance of the estimated models. We repeat this random split 10 times to obtain 10 performance measures for prediction. For the performance measure, we use the Root Mean Square Error (RMSE) for regression datasets and the Area Under the ROC curve (AUROC) for classification datasets.

Table 4: **Prediction performance on high-dimensional datasets.** We report the averages (standard deviations) of the prediction performance for 10 randomly sampled training data from the high-dimensional datasets. The bold faces highlight the best results.

| | | $GA^1M$ | | | $GA^2M$ | | |
|---|---|---|---|---|---|---|---|
| Dataset | Measure | ANOVA $T^1PNN$ | $NA^1M$ | $NB^1M$ | NID + ANOVA $T^2PNN$ | NID + $NA^2M$ | NID + $NB^2M$ |
| MICROSOFT (Qin & Liu, 2013) | RMSE ↓ | 0.756 (0.001) | 0.774 (0.001) | 0.770 (0.001) | **0.754** (0.001) | 0.761 (0.001) | 0.755 (0.001) |
| YAHOO (Yahoo, 2010) | RMSE ↓ | 0.787 (0.002) | 0.797 (0.002) | 0.783 (0.002) | **0.779** (0.001) | 0.793 (0.002) | **0.779** (0.002) |
| MADELON (Guyon, 2004) | AUROC ↑ | 0.587 (0.02) | 0.587 (0.02) | 0.582 (0.03) | **0.605** (0.01) | 0.568 (0.03) | 0.594 (0.02) |

Table 5: **Stability scores on the high-dimensional datasets.** For each dataset, stability scores of of $GA^1M$ ($GA^2M$) models are presented. Lower stability scores imply more stable interpretation. The bold faces highlight the best results.

| | $GA^1M$ | | | $GA^2M$ | | |
|---|---|---|---|---|---|---|
| Dataset | ANOVA $T^1PNN$ | $NA^1M$ | $NB^1M$ | NID + ANOVA $T^2PNN$ | NID + $NA^2M$ | NID + $NB^2M$ |
| MICROSOFT | **0.030** | 0.089 | 0.118 | **0.040** | 0.083 | 0.089 |
| YAHOO | **0.049** | 0.088 | 0.126 | **0.049** | 0.090 | 0.074 |
| MADELON | **0.070** | 0.137 | 0.141 | **0.076** | 0.090 | 0.086 |

Table 3 presents the results of prediction performance of ANOVA-TPNN, NODE-GAM, NAM, and NBM as well as two black box models including deep neural networks (DNN, Rosenblatt (1958)) and XGB (Chen & Guestrin, 2016)). At the final line, the average ranks of each model over the nine datasets are presented, which shows that ANOVA-$T^2$PNN exhibits comparable or superior prediction performance compared to the baseline models. Details about the experiments are given in Appendix B.2.

### 4.4. Application to high-dimensional data

To see whether ANOVA-TPNN is applicable to high-dimensional data, we analyze three additional datasets with input dimensions ranging from 136 to 699. See Table 13 of Appendix for details of these three datasets. For ANOVA-$T^1$PNN, we include all main effects into the model. For ANOVA-$T^2$PNN, however, the number of second order interactions is too large so that considering all the main effects and second order interactions would be difficult unless very large computing resources are available. A simple alternative is to screen out unnecessary second order interactions a priori and include only selected second order interactions (and all the main effects) into the model. In the experiment, we use Neural Interaction Detection (NID) proposed by Tsang et al. (2017) for the interaction screening. The numbers of selected interactions are given in Appendix B.2.

From Table 4 and Table 5, we observe that ANOVA-TPNN shows favorable prediction performance compared to NAM and NBM and estimates the components more stably on high-dimensional datasets. Note that the reported RMSE of NB$^2$M with all second order interactions on MICROSOFT by Radenovic et al. (2022) is 0.750, which indicates that screening interactions using NID does not hamper prediction performance much.

Table 6: **Results of the prediction performance and stability scores of ANOVA-TPNN and Spline-GAM on the CALHOUSING dataset.** The optimal results are highlighted in **bold**.

| Measure | ANOVA-$T^1$PNN | Spline-$GA^1M$ |
|---|---|---|
| RMSE ↓ (std) | **0.614** (0.01) | 0.636 (0.03) |
| Stability score ↓ | **0.012** | 0.033 |

| Measure | ANOVA-$T^2$PNN | Spline-$GA^2M$ |
|---|---|---|
| RMSE ↓ (std) | **0.512** (0.01) | 1.326 (2.058) |
| Stability score ↓ | **0.035** | 0.052 |

### 4.5. Comparison between ANOVA-TPNN and the basis expansion approach

In this section, we conduct experiments to compare ANOVA-TPNN and Spline-GAM which estimates each component by using cubic B-spline basis functions. We evaluate the prediction performance and stability of the component estimation in ANOVA-TPNN and Spline-GAM on CALHOUSING dataset. We implement Spline-GAM using pygam python package (Servén & Brummitt, 2018).

Table 6 presents the results of the prediction performance and stability scores of ANOVA-TPNN and Spline-GAM on CALHOUSING dataset. A surprising result is that ANOVA-TPNN is superior to Spline-GAM in both of prediction performance and stability of component estimation. By investigating details of the empirical results, we find that Spline-GAM is vulnerable to input outliers. That is, when there is an outlier input (i.e. an input vector in the test data locating outside the range of input vectors in the training data), the prediction at the outlier could be inaccurate since the B-splie basis uses the linear extrapolation outside the domain of the training data. In contrast, the basis neural networks in TPNN use the sigmoid activation function which is bounded outside the range of input vectors and so robust to input outliers. The details of experimental results for Spline-GAM are presented in Appendix N.

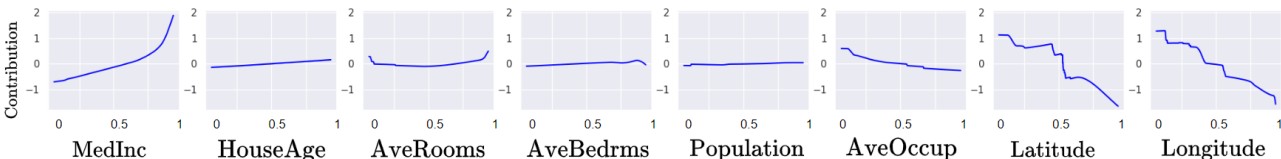

Figure 1: **Plots of the functional relations of the main effects in ANOVA-T$^1$PNN on CALHOUSING dataset.**

Table 7: **Importance scores of ANOVA-T$^1$PNN on CALHOUSING dataset.**

| Feature index | 7 | 8 | 1 | 6 | 3 | 2 | 4 | 5 |
|---|---|---|---|---|---|---|---|---|
| Importance score | 1.000 | 0.906 | 0.564 | 0.284 | 0.107 | 0.093 | 0.057 | 0.049 |

Table 8: **Importance scores of ANOVA-T$^2$PNN on CALHOUSING dataset.**

| Feature index | 6 | (7,8) | (1,7) | (3,8) | 7 | (1,8) | (4,8) | (2,7) | (1,5) | 8 |
|---|---|---|---|---|---|---|---|---|---|---|
| Importance score | 1.000 | 0.347 | 0.324 | 0.268 | 0.258 | 0.247 | 0.212 | 0.194 | 0.193 | 0.178 |

## 4.6. Experiment results for NBM-TPNN

We conduct experiment to evaluate NBM-TPNN. Table 9 shows the prediction performance and stability score of NBM-T$^1$PNN on CALHOUSING dataset and ABALONE dataset. We observe that NBM-T$^1$PNN also exhibits similar prediction performance and stability to ANOVA-T$^1$PNN. The plots of estimated main effects are given in Appendix F and the results of computation time are given in Appendix J. We can observe that relative computation times of NBM-TPNN compared to ANOVA-TPNN decreases as the number of input features increases.

Table 9: **Results of prediction performance and stability scores of ANOVA-T$^1$PNN and NBM-T$^1$PNN.**

| Dataset | Measure | ANOVA-T$^1$PNN | NBM-T$^1$PNN |
|---|---|---|---|
| CALHOUSING | RMSE ↓ (std) | 0.614 (0.001) | 0.604 (0.001) |
| | Stability score ↓ | 0.012 | 0.009 |
| WINE | RMSE ↓ (std) | 0.725 (0.02) | 0.720 (0.02) |
| | Stability score ↓ | 0.011 | 0.017 |

## 4.7. Interpretability of ANOVA-TPNN

We consider the two concepts of interpretation: Local and Global which are roughly defined as:

**Local Interpretation:** Information about how each feature of a given datum affects the prediction. SHAP (Lundberg, 2017) is a notable example of local interpretation. For the functional ANOVA model, the predictive values of each component at a given datum would be considered as local interpretation.

**Global Interpretation:** Information about how each feature is related to the final prediction model. The importance scores of each feature (e.g. global SHAP (Molnar, 2020)) and the functional relations between each feature and the prediction model (e.g. the dependency plot of SHAP (Molnar, 2020) are examples of global interpretation. For the functional ANOVA model, the importance score, which can be defined as the $l_1$ norm of the corresponding component as

is done in Section 4.2, and the functional relation identified by the functional form of each component are two tools for global interpretation.

### 4.7.1. ILLUSTRATION OF INTERPRETABILITY ON CALHOUSING DATASET.

Table 10: **Feature descriptions of CALHOUSING dataset.**

| Feature name | Index | Description | Feature type |
|---|---|---|---|
| MedInc | 1 | Median income in block | Numerical |
| HouseAge | 2 | Median house age in block | Numerical |
| AveRooms | 3 | Average number of rooms | Numerical |
| AveBedrms | 4 | Average number of bedrooms | Numerical |
| Population | 5 | Population in block | Numerical |
| AveOccup | 6 | Average house occupancy | Numerical |
| Latitude | 7 | Latitude of house block | Numerical |
| Longitude | 8 | Longitude of house block | Numerical |

**Local Interpretation on CALHOUSING dataset.** We conduct an experiment on CALHOUSING (Pedregosa et al., 2011a) dataset to illustrate local interpretation of ANOVA-T$^1$PNN. Note that ANOVA-T$^1$PNN is given as

$$\hat{f}_{\text{ANOVA-T}^1\text{PNN}}(\mathbf{x}) = \sum_{j=1}^{8} \hat{f}_j(x_j).$$

Thus, it is reasonable to treat $\hat{f}_j(x_j)$ as the contribution of $x_j$ to $\hat{f}(\mathbf{x})$. In fact, we have seen in Section 3.1 that this contribution is equal to SHAP (Lundberg, 2017). As an illustration, for a given datum

$$\mathbf{x} = (-0.2378, -0.4450, 0.0036, -0.1531,$$
$$0.3814, -0.067, 0.5541, -0.1111)^\top,$$

the contributions of each feature to $\hat{f}(\mathbf{x})$ are

$$(\hat{f}_1, ..., \hat{f}_8) = (-4.9900, 0.3278, -0.0456, 0.4432,$$
$$-0.1730, 2.7521, -11.6190, 6.5184).$$

That is, the 7th variable contributes most to the prediction value of $\hat{f}(\mathbf{x})$, which can be interpreted as 'the housing price is low because the latitude is not good'.

**Global Interpretation on CALHOUSING dataset.** Figure 1 and Table 7 present the functional relations of each input feature to the prediction model learned by ANOVA-T$^1$PNN and their importance scores. From these results, we can see that the location is the most important features and the south-west area is the most expensive.

Table 8 describes the 10 most important components with descending order of the importance scores of ANOVA-T$^2$PNN normalized by the maximum importance score. The results are bit different from those of ANOVA-T$^1$PNN. In particular, the interaction between 'latitude' and 'longitude' emerges as a new important feature while the main effects of 'latitude' and 'longitude' become less important.

### 4.8. ANOVA-TPNN with monotone constraints

**Monotone constraint.** In practice, prior knowledge that some main effects are monotone functions is available and it is needed to reflect this prior knowledge in the training phase. A notable example is the credit scoring model where certain input features should have monotone main effects (Chen & Li, 2014; Chen & Ye, 2022).

An additional advantage of ANOVA-TPNN is to accommodate the monotone constraints in the model easily. Suppose that $f_j$ is monotonically increasing. Then, ANOVA-TPNN can estimate $f_j$ monotonically increasingly by letting the $\beta_{jk}$ in equation (3) be less than or equal to 0.

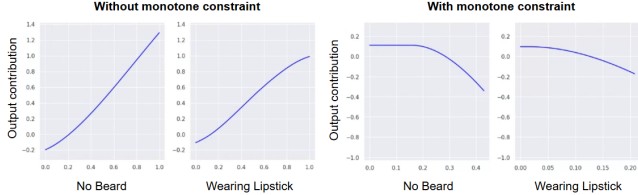

Figure 2: **Plots of the functional relations of 'No Beard' and 'Wearing Lipstick' on CELEBA dataset estimated by ANOVA-T$^1$PNN with and without the monotone constraint.**

**Application to Image data.** Monotone constraint helps avoiding unreasonable interpretation. To illustrate this advantage, we conduct an experiment with an image dataset. We use CELEBA (Liu et al., 2015) dataset which has 40 binary attributes for each image. To apply ANOVA-TPNN to CELEBA dataset, we consider Concept Bottleneck Model (CBM, Koh et al. (2020)) similar to the one used in Radenovic et al. (2022). In CBM, rather than directly inputting the embedding vector derived from an image data through a CNN into a classifier, the CNN initially predicts each concept accompanied with each image. Then, these predicted values of each concept are subsequently used as the input of a DNN classifier. For our experiment, we use a pretrained ResNet18, where the last layer consists of a linear transformation with a softmax activation function, and we replace the final DNN classifier with ANOVA-TPNN.

Among the attributes, we set 'gender' as the target label and the remaining attributes are set as concepts for images. Since 'male' is labeled as 1 and 'female' as 0, a higher value of each component results in a higher chance of being classified as 'male'.

Figure 2 presents the functional relations of the two main effects corresponding to the concepts 'No Beard' and 'Wearing Lipstick', estimated on a randomly sampled training dataset with and without the monotone constraint. Note that the functional relations are quite different even though their prediction performances, which are given in Table 18 of Appendix D, are similar. It is a common sense that an image having the concept of 'No Beard' and 'Wearing Lipstick' has a higher chance of being a female and thus the functional relations are expected to be decrease. Figure 2 illustrates that a completely opposite result to our common sense could be obtained in practice. Implications of the opposite functional relations to interpretation of each image are discussed in Appendix D.

### 4.9. Additional Experiments

In Appendix K, we explore ANOVA-TPNN with ReLU activation function. In Appendix L, we confirm that the component estimation of ANOVA-T$^2$PNN becomes highly unstable when the sum-to-zero condition is not imposed. Finally, in Appendix M, we discuss a method for enforcing the sum-to-zero condition after training NAM or NBM.

## 5. Conclusion

In this paper, we propose a novel XAI model called ANOVA-TPNN for estimating the functional ANOVA model stably by use of TPNNs. We theoretically demonstrate that ANOVA-TPNN can approximate a smooth function well. We also empirically show that prediction performance of ANOVA-TPNN is comparable to its competitors. Additionally, we proposed NBM-TPNN that improves the scalability of ANOVA-TPNN using the idea from NBM (Radenovic et al., 2022). One advantage of the basis function in NBM-TPNN is that the number of basis functions does not depend on the dimension of the input feature.

Even though it is computationally more efficient that standard basis expansion approaches, ANOVA-TPNN with high order interactions is still computationally demanding to be applied to high-dimensional data. A remedy would be to develop a tool to select and estimate signal components simultaneously in the functional ANOVA model, which makes it possible to detect higher order interactions in high-dimensional data.

## Acknowledgements

This research was partly supported by Institute of Information & communications Technology Planning & Evaluation (IITP) grant funded by the Korea government(MSIT) (No.RS-2022-II220184, Development and Study of AI Technologies to Inexpensively Conform to Evolving Policy on Ethics), the National Research Foundation of Korea(NRF) grant funded by the Korea government(MSIT) (RS-2025-00556079), the Institute of Information & communications Technology Planning & Evaluation (IITP) grant funded by the Korea government(MSIT) [NO.RS-2021-II211343, Artificial Intelligence Graduate School Program (Seoul National University)], and Samsung Electronics Co., Ltd.

## Impact Statement

This paper presents work whose goal is to advance the field of Machine Learning. There are many potential societal consequences of our work, none which we feel must be specifically highlighted here.

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

# Supplementary material

# Appendix

## A. Proof of Theorem 3.3

**A.1. Case of** $|S| = 1$

Without loss of generality, we assume that $S = \{j\}$. The proof is done by the following 2 steps.

- Firstly, we find a function $f_{\mathcal{E},j}$ that approximates the true component $g_{0,j}$ well.

- Next, we decompose the approximation function $f_{\mathcal{E},j}$ into a sum of TPNNs.

**Step 1.** Finding a approximation function $f_{\mathcal{E},j}$

Let $0 < p_L < p_R < \infty$ be the lower and upper bounds of the density of $\mu_j$, respectively. Let $\{\Omega_k^j\}_{k=1}^K = \{[\chi_{k-1}^j, \chi_k^j)\}_{k=1}^K$ be an interval partition of $\mathcal{X}_j$ such that $\mu_j(\Omega_k^j) = \frac{1}{K}$ for some positive integer $K$. Then, we have $|\chi_k^j - \chi_{k-1}^j| \le \frac{1}{p_L K}$ for $k \in [K]$. For $\gamma_j = 1/K^3$, we define $\ell_{j,k}(\cdot)$ as

$$\ell_{j,1}(x) = 1 - \sigma\left(\frac{x - \chi_1^j}{\gamma_j}\right),$$

$$\ell_{j,k}(x) = \sigma\left(\frac{x - \chi_{k-1}^j}{\gamma_j}\right) - \sigma\left(\frac{x - \chi_k^j}{\gamma_j}\right), \qquad k \in \{2, \dots, K-1\}$$

$$\ell_{j,K}(x) = \sigma\left(\frac{x - \chi_{K-1}^j}{\gamma_j}\right).$$

Note that for every $x \in \mathcal{X}_j$, $\sum_{k=1}^K \ell_k^j(x) = 1$ and for every $k \in [K]$, $0 \le \ell_{j,k}(\cdot) \le 1$ holds. Also, $\{\ell_{j,k}\}_{k=1}^K$ satisfy Lemma A.1 below, whose proof is provdied in Section A.2.

**Lemma A.1.** *For any $k \in [K]$, we have*

$$\mathbb{E}_{\mathcal{X}_j}[\ell_{j,k}(X_j)\mathbb{I}(X_j \notin \Omega_k)] \le \frac{2p_U}{K^2}$$

*and*

$$\mathbb{E}_{\mathcal{X}_j}[\ell_{j,k}(X_j)\mathbb{I}(X_j \in \Omega_k)] \ge \frac{C_0}{K}$$

*for some positive constant $C_0$.*

Now, we consider the function defined as

$$f_{\mathcal{E},j}(x) = \sum_{k=1}^K \delta_k \ell_{j,k}(x),$$

where $\delta_k$s are defined as

$$\delta_k = \frac{\mathbb{E}_{\mathcal{X}_j}[\ell_{j,k}(X_j)g_{0,j}(X_j)]}{\mathbb{E}_{\mathcal{X}_j}[\ell_{j,k}(X_j)]}$$

for $k \in [K]$. Then, for any $x \in \mathcal{X}_j$, we have

$$
\begin{aligned}
\left| g_{0,j}(x) - f_{\mathcal{E},j}(x) \right| &= \left| g_{0,j}(x) - \sum_{k=1}^{K} \delta_k \ell_{j,k}(x) \right| \\
&= \left| \sum_{k=1}^{K} (g_{0,j}(x) - \delta_k) \ell_{j,k}(x) \right| \\
&\leq \sum_{k=1}^{K} |g_{0,j}(x) - \delta_k| \ell_{j,k}(x) \\
&= \sum_{k=1}^{K} \left| g_{0,j}(x) - \frac{\mathbb{E}_{\mathcal{X}_j}[\ell_{j,k}(X_j) g_{0,j}(X_j)]}{\mathbb{E}_{\mathcal{X}_j}[\ell_{j,k}(X_j)]} \right| \ell_{j,k}(x) \\
&= \sum_{k=1}^{K} \left| \frac{\mathbb{E}_{\mathcal{X}_j}[\ell_{j,k}(X_j) g_{0,j}(x)] - \mathbb{E}_{\mathcal{X}_j}[g_{0,j}(X_j) \ell_{j,k}(X_j)]}{\mathbb{E}_{\mathcal{X}_j}[\ell_{j,k}(X_j)]} \right| \ell_{j,k}(x) \\
&\leq L \sum_{k=1}^{K} \left| \frac{\mathbb{E}_{\mathcal{X}_j}[\ell_{j,k}(X_j)|x - X_j|]}{\mathbb{E}_{\mathcal{X}_j}[\ell_{j,k}(X_j)]} \right| \ell_{j,k}(x),
\end{aligned}
\tag{5}
$$

where $L > 0$ is the Lipschitz constant for $g_{0,j}(\cdot)$.

For a given $x \in \mathcal{X}_j$, let $r \in [K]$ be the index such that $x \in [\chi_{r-1}^j, \chi_r^j]$. For $k \in \{r-1, r, r+1\}$, we have

$$
\begin{aligned}
\frac{\mathbb{E}_{\mathcal{X}_j}[\ell_{j,k}(X_j)|x - X_j|]}{\mathbb{E}_{\mathcal{X}_j}[\ell_{j,k}(X_j)]} &\leq \frac{\mathbb{E}_{\mathcal{X}_j}[\ell_{j,k}(X_j)|x - X_j|\mathbb{I}(X_j \in \Omega_k^j)]}{\mathbb{E}_{\mathcal{X}_j}[\ell_{j,k}(X_j)\mathbb{I}(X_j \in \Omega_k^j)]} + \frac{\mathbb{E}_{\mathcal{X}_j}[\ell_{j,k}(X_j)|x - X_j|\mathbb{I}(X_j \notin \Omega_k^j)]}{\mathbb{E}_{\mathcal{X}_j}[\ell_{j,k}(X_j)\mathbb{I}(X_j \in \Omega_k^j)]} \\
&\leq \frac{\mathbb{E}_{\mathcal{X}_j}[\ell_k(X_j)(\frac{2}{p_L K})\mathbb{I}(X_j \in \Omega_k^j)]}{\mathbb{E}_{\mathcal{X}_j}[\ell_{j,k}(X_j)\mathbb{I}(X_j \in \Omega_k^j)]} + \frac{2\mathbb{E}_{\mathcal{X}_j}[\ell_{j,k}(X_j)\mathbb{I}(X_j \notin \Omega_k^j)]}{\mathbb{E}_{\mathcal{X}_j}[\ell_{j,k}(X_j)\mathbb{I}(X_j \in \Omega_k^j)]} \\
&\leq \frac{2}{p_L K} + \frac{4p_U}{C_0 K} \\
&= \frac{C'}{K}
\end{aligned}
$$

for $C' = \frac{2}{p_L} + \frac{4p_U}{C_0}$. Also, for $k \geq r + 2$, we have $x \leq \chi_r^j$ and $\chi_{k-1}^j \geq \chi_{r+1}^j$, and hence

$$
\begin{aligned}
|\ell_{j,k}(x)| &\leq \sigma\left( \frac{x - \chi_{k-1}^j}{\gamma_j} \right) \\
&\leq \sigma\left( \frac{\chi_r^j - \chi_{r+1}^j}{\gamma_j} \right) \\
&\leq \frac{1}{1 + \exp(K)}.
\end{aligned}
$$

To sum up, we have

$$
\begin{aligned}
|g_{0,j}(x) - f_{\mathcal{E},j}(x)| &\leq L \sum_{k=1}^{K} \left| \frac{\mathbb{E}_{\mathcal{X}_j}[\ell_{j,k}(X_j)|x - X_j|]}{\mathbb{E}_{\mathcal{X}_j}[\ell_{j,k}(X_j)]} \right| \ell_{j,k}(x) \\
&= L\left( \sum_{k \in \{r-1,r,r+1\}} \left| \frac{\mathbb{E}_{\mathcal{X}_j}[\ell_{j,k}(X_j)|x - X_j|]}{\mathbb{E}_{\mathcal{X}_j}[\ell_{j,k}(X_j)]} \right| \ell_{j,k}(x) + \sum_{k \notin \{r-1,r,r+1\}} \left| \frac{\mathbb{E}_{\mathcal{X}_j}[\ell_{j,k}(X_j)|x - X_j|]}{\mathbb{E}_{\mathcal{X}_j}[\ell_{j,k}(X_j)]} \right| \ell_{j,k}(x) \right) \\
&\leq L\left( \frac{3C'}{K} + \frac{1}{1 + \exp(K)} \right).
\end{aligned}
$$

**Step 2.** Decomposing the approximation function $f_{\mathcal{E},j}$ into a sum of TPNNs.

Now, for $f_{\mathcal{E},j}(x) = \sum_{k=1}^{K} \delta_k \ell_{j,k}(x)$, our goal is to show that there exists $\beta_{j,k}$ and $\phi_j(\cdot|\theta_{j,k})$, $k = 1, ..., K-1$ such that

$$f_{\mathcal{E},j}(x) = \sum_{k=1}^{K-1} \beta_{j,k}\phi_j(x|\phi_{j,k})$$

for every $x \in \mathcal{X}_j$. To derive this result, we use Lemma A.2 below whose proof is provided in Appendix A.2.

**Lemma A.2.** *For every $T \in \{2, \ldots, K\}$, we define*

$$\ell_{1,T}(x) = 1 - \sigma\left(\frac{x - \chi_1^j}{\gamma_j}\right),$$

$$\ell_{t,T}(x) = \sigma\left(\frac{x - \chi_{t-1}^j}{\gamma_j}\right) - \sigma\left(\frac{x - \chi_t^j}{\gamma_j}\right), \qquad t \in \{2, \ldots, T-1\}$$

$$\ell_{T,T}(x) = \sigma\left(\frac{x - \chi_{T-1}^j}{\gamma_j}\right).$$

*Then, for any given $T \in \{3, \ldots, K\}$ and for any $\rho_{1,T}, \ldots, \rho_{T,T}$ satisfying*

$$\mathbb{E}_{\mathcal{X}_j}\left[\sum_{t=1}^{T} \rho_{t,T}\ell_{t,T}(X_j)\right] = 0, \tag{6}$$

*there exist $\rho_{1,T-1}, \ldots, \rho_{T-1,T-1}$, $\eta$ and $\tau$ such that*

$$\sum_{t=1}^{T} \rho_{t,T}\ell_{t,T}(x) = \sum_{t=1}^{T-1} \rho_{t,T-1}\ell_{t,T-1}(x) + \left[\eta \cdot \left(1 - \sigma\left(\frac{x - \chi_{T-1}^j}{\gamma_j}\right)\right) + \tau \cdot \sigma\left(\frac{x - \chi_{T-1}^j}{\gamma_j}\right)\right], \tag{7}$$

$$\tag{8}$$

*and*

$$\mathbb{E}_{\mathcal{X}_j}\left[\sum_{t=1}^{T-1} \rho_{t,T-1}\ell_{t,T-1}(X_j)\right] = 0, \quad \mathbb{E}_{\mathcal{X}_j}\left[\eta \cdot \left(1 - \sigma\left(\frac{X_j - \chi_{T-1}^j}{\gamma_j}\right)\right) + \tau \cdot \sigma\left(\frac{X_j - \chi_{T-1}^j}{\gamma_j}\right)\right] = 0. \tag{9}$$

Since

$$\begin{aligned}
\mathbb{E}_{\mathcal{X}_j}[f_{\mathcal{E},j}(X_j)] &= \mathbb{E}_{\mathcal{X}_j}\left[\sum_{k=1}^{K} \delta_k \ell_{j,k}(X_j)\right] \\
&= \sum_{k=1}^{K} \frac{\mathbb{E}_{\mathcal{X}_j}[\ell_{j,k}(X_j)g_{0,j}(X_j)]}{\mathbb{E}_{\mathcal{X}_j}[\ell_{j,k}(X_j)]}\mathbb{E}_{\mathcal{X}_j}[\ell_{j,k}(X_j)] \\
&= \sum_{k=1}^{K} \mathbb{E}_{\mathcal{X}_j}[\ell_{j,k}(X_j)g_{0,j}(X_j)] \\
&= \mathbb{E}_{\mathcal{X}_j}\left[\sum_{k=1}^{K} \ell_{j,k}(X_j)g_{0,j}(X_j)\right] \\
&= \mathbb{E}_{\mathcal{X}_j}[g_{0,j}(X_j)] \\
&= 0,
\end{aligned}$$

we can decompose $f_{\mathcal{E},j}(\cdot)$ into the sum of functions using Lemma A.2 and mathematical induction, i.e.,

$$f_{\mathcal{E},j}(x) = (\rho_{1,2}\ell_{1,2}(x) + \rho_{2,2}\ell_{2,2}(x)) + \sum_{k=2}^{K-1} \left(\eta_k\left(1 - \sigma\left(\frac{X_j - \chi_k^j}{\gamma_j}\right)\right) + \tau_k\sigma\left(\frac{X_j - \chi_k^j}{\gamma_j}\right)\right).$$

Here, since

$$\mathbb{E}_{\mathcal{X}_j}[\rho_{1,2}\ell_{1,2}(X_j) + \rho_{2,2}\ell_{2,2}(X_j)] = 0,$$

we can express $\rho_{1,2}\ell_{1,2}(x) + \rho_{2,2}\ell_{2,2}(x)$ using $\beta_{j,1}$ and $\theta_{j,1}$ such that

$$\rho_{1,2}\ell_{1,2}(x) + \rho_{2,2}\ell_{2,2}(x) = \beta_{j,1}\phi_j(x|\theta_{j,1}).$$

Similarly, since

$$\mathbb{E}_{\mathcal{X}_j}\left[\eta_k\left(1 - \sigma\left(\frac{X_j - \chi_k^j}{\gamma_j}\right)\right) + \tau_k\sigma\left(\frac{X_j - \chi_k^j}{\gamma_j}\right)\right] = 0$$

we can express $\eta_k\left(1 - \sigma\left(\frac{x-\chi_k^j}{\gamma_j}\right)\right) + \tau_k\sigma\left(\frac{x-\chi_k^j}{\gamma_j}\right)$ using $\beta_{j,k}$ and $\theta_{j,k}$ such that

$$\eta_k\left(1 - \sigma\left(\frac{x - \chi_k^j}{\gamma_j}\right)\right) + \tau_k\sigma\left(\frac{x - \chi_k^j}{\gamma_j}\right) = \beta_{j,k}\phi_j(x|\theta_{j,k})$$

for $k = 2, ..., K - 1$.

**Completion of the proof.** By summarizing the results obtained in Step 1 and Step 2 and letting $K_j = K - 1$, we have

$$\left\|g_{0,j}(\cdot) - \sum_{k=1}^{K_j}\beta_{j,k}\phi_j(\cdot|\theta_{j,k}) = f_{\mathcal{E}}(\cdot)\right\|_\infty \leq L\left(\frac{3C'}{K} + \frac{1}{1 + \exp(K)}\right).$$

$$\leq \frac{C_j}{K_j + 1}$$

where $C_j$ is a positive constant. $\square$

### A.2. Proof of Lemma A.1

For $x \leq \chi_{k-1}^j - \frac{1}{K^2}$, we have

$$
\begin{aligned}
|\ell_{j,k}(x)| &\leq \sigma\left(\frac{x - \chi_{k-1}^j}{\gamma_j}\right) \\
&\leq \sigma\left(-\frac{1}{K^2\gamma_j}\right) \\
&= \frac{1}{1 + \exp(K)}.
\end{aligned}
$$

Also, for $x \geq \chi_k^j + \frac{1}{K^2}$, we have

$$
\begin{aligned}
|\ell_{j,k}(x)| &\leq 1 - \sigma\left(\frac{x - \chi_k^j}{\gamma_j}\right) \\
&\leq 1 - \sigma\left(\frac{1}{K^2\gamma_j}\right) \\
&= \frac{1}{1 + \exp(K)}.
\end{aligned}
$$

Hence, we obtain

$$
\begin{aligned}
\mathbb{E}_{\mathcal{X}_j}[\ell_{j,k}(X_j)\mathbb{I}(X_j \notin \Omega_k^j)] &\leq \mathbb{P}\left(X_j \in \left[\chi_{k-1}^j - \frac{1}{K^2}, \chi_{k-1}^j\right] \bigcup \left[\chi_k^j, \chi_k^j + \frac{1}{K^2}\right]\right) + \frac{1}{1 + \exp(K)} \\
&\leq \frac{2p_U}{K^2}.
\end{aligned}
$$

Also, for $x \in \left[\chi_{k-1}^j + \frac{1}{K^2}, \chi_k^j - \frac{1}{K^2}\right]$, we have

$$
\begin{aligned}
\ell_{j,k}(x) &\geq \sigma\left(\frac{x - \chi_{k-1}^j}{\gamma_j}\right) - \sigma\left(\frac{x - \chi_k^j}{\gamma_j}\right) \\
&\geq \sigma(K) - \sigma(-K) \\
&\geq \frac{1}{2}
\end{aligned}
$$

for sufficiently large $K$. Hence, we obtain

$$
\begin{aligned}
\mathbb{E}_{\mathcal{X}_j}[\ell_{j,k}(X_j)\mathbb{I}(X_j \in \Omega_k^j)] &\geq \mathbb{P}\left(X_j \in \left[\chi_{k-1}^j + \frac{1}{K^2}, \chi_k^j - \frac{1}{K^2}\right]\right) \cdot \frac{1}{2} \\
&\geq \frac{C_0}{K}
\end{aligned}
$$

for some positive constant $C_0$. $\qquad\square$

### A.3. Proof of Lemma A.2

We define

$$\eta := -\mathbb{E}_{\mathcal{X}_j}[\ell_{T,T}(X_j)](\rho_T - \rho_{T-1}),$$

$$\tau := \sum_{t=1}^{T-1} \mathbb{E}_{\mathcal{X}_j}[\ell_{t,T}(X_j)](\rho_T - \rho_{T-1}),$$

$$\kappa_t := \rho_t - \eta$$

for $t \in [T-1]$. Then, we obtain (7) by

$$\sum_{t=1}^{T-1} \kappa_t \ell_{t,T-1}(x) + \left[ \eta \cdot \left( 1 - \sigma \left( \frac{x - \chi_{T-1}^j}{\gamma_j} \right) \right) + \tau \cdot \sigma \left( \frac{x - \chi_{T-1}^j}{\gamma_j} \right) \right]$$

$$= \sum_{t=1}^{T-2} (\rho_t - \eta) \ell_{t,T}(x) + (\rho_{T-1} - \eta) \cdot \sigma \left( \frac{x - \chi_{T-2}^j}{\gamma_j} \right) + (\tau - \eta) \cdot \sigma \left( \frac{x - \chi_{T-1}^j}{\gamma_j} \right) + \eta$$

$$= \sum_{t=1}^{T-2} (\rho_t - \eta) \ell_{t,T}(x) + (\rho_{T-1} - \eta) \cdot \left( \sigma \left( \frac{x - \chi_{T-2}^j}{\gamma_j} \right) - \sigma \left( \frac{x - \chi_{T-1}^j}{\gamma_j} \right) \right)$$

$$\quad + (\rho_{T-1} - \eta) \cdot \sigma \left( \frac{x - \chi_{T-1}^j}{\gamma_j} \right) + (\rho_T - \rho_{T-1}) \cdot \sigma \left( \frac{x - \chi_{T-1}^j}{\gamma_j} \right) + \eta$$

$$= \sum_{t=1}^{T-1} (\rho_t - \eta) \ell_{t,T}(x) + (\rho_T - \eta) \cdot \sigma \left( \frac{x - \chi_{T-1}^j}{\gamma_j} \right) + \eta$$

$$= \sum_{t=1}^{T} (\rho_t - \eta) \ell_{t,T}(x) + \eta$$

$$= \sum_{t=1}^{T} \rho_t \ell_{t,T}(x).$$

$\square$

**A.4. Case of $|S| = d$**

Without loss of generality, we consider $S = \{1, ..., d\}$. Similarly to the case of $S = \{j\}$, we define an interval partition of $\mathcal{X}_j$ as $\{\Omega_k^j\}_{k=1}^K = \{[\chi_{k-1}^j, \chi_k^j]\}_{k=1}^K$ such that $\mu_j(\Omega_k^j) = \frac{1}{K}$ and $|\chi_{k-1}^j - \chi_k^j| \leq \frac{1}{p_L K}$ for $j = 1, ..., d$. Additionally, let $\Omega_{k_1,...,k_d} = \Omega_{k_1}^1 \times \cdots \times \Omega_{k_d}^d$ for $k_j \in [K]$, $j = 1, ..., d$. For $\Omega_{k_1,...,k_d}$, we define $\ell_{k_1,...,k_d}(\mathbf{x}_S)$ as

$$\ell_{k_1,...,k_d}(\mathbf{x}_S) = \prod_{j=1}^d \ell_{j,k_j}(x_j),$$

where $\ell_{j,k_j}(\cdot)$ is defined in the same way as in Step 1 for $|S| = 1$.

Consider the approximation function $f_{\mathcal{E},S}(\mathbf{x}_S)$ defined as

$$f_{\mathcal{E},S}(\mathbf{x}_S) = \sum_{k_1,...,k_d=1}^K \delta_{k_1,...,k_d} \ell_{k_1,...,k_d}(\mathbf{x}_S), \tag{10}$$

where

$$\delta_{k_1,...,k_d} = \frac{\int_{\mathcal{X}_S} \ell_{k_1,...,k_d}(\mathbf{x}_S) g_{0,S}(\mathbf{x}_S) \mathbb{P}_S(d\mathbf{x}_S)}{\int_{\mathcal{X}_S} \ell_{k_1,...,k_d}(\mathbf{x}_S) \mathbb{P}_S(d\mathbf{x}_S)}.$$

Here, $\mathbb{P}_S = \prod_{j \in S} \mathbb{P}_j$, where $\mathbb{P}_j$ is the population distribution of $X_j$. Using a similar approach as is done in Step 1 for $|S| = 1$, we have

$$\left| g_{0,S}(\mathbf{x}_S) - f_{\mathcal{E},S}(\mathbf{x}_S) \right| \leq \frac{C_S d}{K}$$

for some positive constant $C_S$.

In turn, we decompose $f_{\mathcal{E},S}$ by Lemma A.3, i.e., there exist $\beta_{S,k}$ and $\theta_{S,k}$ for $k = 1, ..., (K-1)^d$ such that

$$f_{\mathcal{E},S}(\mathbf{x}_S) = \sum_{k_1,...,k_d=1}^K \delta_{k_1,...,k_d} \ell_{k_1,...,k_d}(\mathbf{x}_S)$$

$$= \sum_{k=1}^{(K-1)^d} \beta_{S,k} \phi_S(\mathbf{x}_S | \theta_{S,k}).$$

Finally, we have

$$\left\| g_{0,S}(\cdot) - \sum_{k=1}^{K_S} \beta_{S,k} \phi_S(\cdot | \theta_{S,k}) \right\|_\infty \leq \frac{C_S d}{K_S^{\frac{1}{d}} + 1}$$

where $K_S = (K-1)^d$ and $C_S$ is a positive constant. $\qquad \square$

**Lemma A.3.** *For a given $f_{\mathcal{E},S}(\boldsymbol{x}_S) = \sum_{k_1,\ldots,k_{|S|}=1}^{K} \delta_{k_1,\ldots,k_{|S|}} \ell_{k_1,\ldots,k_{|S|}}(\boldsymbol{x}_S)$ defined in (10), there exist $\beta_{S,k}$ and $\theta_{S,k}$ for $k = 1, \ldots, (K-1)^{|S|}$ such that*

$$f_{\mathcal{E},S}(\boldsymbol{x}_S) = \sum_{k=1}^{(K-1)^{|S|}} \beta_{S,k} \phi_S(\boldsymbol{x}_S|\theta_{S,k}).$$

Proof.)

Without loss of generality, we assume that $S = \{1, \ldots, d\}$. Let $\mathbf{part}(f)_j$ be an interval partition for $x_j$ used when the function $f$ is defined. For example, for $f_{\mathcal{E},S}(\mathbf{x}_S)$, we have

$$\mathbf{part}(f_{\mathcal{E},S})_j = \{[\chi_{k-1}^j, \chi_k^j]\}_{k=1}^K \quad , \quad \left|\mathbf{part}(f_{\mathcal{E},S})_j\right| = K$$

for $j = 1, \ldots, d$. Since, $\mathbb{E}_{\mathcal{X}_1}[f_{\mathcal{E},S}(\mathbf{X}_S)] = 0$, as is done in the proof of Step 1 for $|S| = 1$, by Lemma A.2 and mathematical induction, we decompose $f_{\mathcal{E},S}(\cdot)$ into a sum of $f_1^{(1)}(\cdot), \ldots, f_{K-1}^{(1)}(\cdot)$ such that

$$f_{\mathcal{E},S}(\mathbf{x}_S) = \sum_{k_1=1}^{K-1} f_{k_1}^{(1)}(\mathbf{x}_S),$$

where $f_{k_1}^{(1)}(\cdot)$s satisfy the sum-to-zero condition, $\mathbf{part}(f_{k_1}^{(1)})_1 = 2$ and $\mathbf{part}(f_{k_1}^{(1)})_2 = \cdots = \mathbf{part}(f_{k_1}^{(1)})_d = K$ for any $k_1 \in [K-1]$. Similarly, for any $k_1 \in [K-1]$, we decompose $f_{k_1}^{(1)}(\cdot)$ into a sum of $f_{k_1,1}^{(1,2)}(\cdot), \ldots, f_{k_1,K-1}^{(1,2)}(\cdot)$ such that

$$f_{k_1}^{(1)}(\cdot) = \sum_{k_2=1}^{K-1} f_{k_1,k_2}^{(1,2)}(\cdot),$$

where $f_{k_1,k_2}^{(1,2)}(\cdot)$s satisfy the sum-to-zero condition, $\mathbf{part}(f_{k_1,k_2}^{(1,2)})_1 = \mathbf{part}(f_{k_1,k_2}^{(1,2)})_2 = 2$ and $\mathbf{part}(f_{k_1,k_2}^{(1,2)})_j = K$ for $j = 3, \ldots, d$ and $k_2 = 1, \ldots, K-1$. We repeat this decomposition to have $\{f_{k_1,\ldots,k_d}^{1,\ldots,d}\}_{k_1,\ldots,k_d=1}^{K}$ such that

$$f_{\mathcal{E},S}(\cdot) = \sum_{k_1=1}^{K-1} \cdots \sum_{k_d=1}^{K-1} f_{k_1,\ldots,k_d}^{(1,\ldots,d)}(\cdot),$$

where $f_{k_1,\ldots,k_d}^{(1,\ldots,d)}(\cdot)$s satisfy the sum-to-zero condition, $\mathbf{part}(f_{k_1,\ldots,k_d}^{(1,\ldots,d)})_j = 2$ for $j \in [d]$ and $k_i \in [K-1]$ for all $i \in [d]$.

For given $i \in [d]$ and $k_i \in [K-1]$, we can express $f_{k_1,\ldots,k_d}^{(1,\ldots,d)}(\cdot)$ using $\{\eta_{k_1,\ldots,k_d}^j, \tau_{k_1,\ldots,k_d}^j\}_{j=1}^d, \gamma_S$, and $\{\chi_{k_j}^j\}_{j=1}^d$ such that

$$f_{k_1,\ldots,k_d}^{(1,\ldots,d)}(\mathbf{x}_S) = \prod_{j=1}^{d} \left\{ \eta_{k_1,\ldots,k_d}^j \left(1 - \sigma\left(\frac{x_j - \chi_{k_j}^j}{\gamma_S}\right)\right) + \tau_{k_1,\ldots,k_d}^j \sigma\left(\frac{x_j - \chi_{k_j}^j}{\gamma_S}\right) \right\}. \tag{11}$$

Since $\prod_{j=1}^{d} \left\{ \eta_{k_1,\ldots,k_d}^j \left(1 - \sigma\left(\frac{x_j - \chi_{k_j}^j}{\gamma_S}\right)\right) + \tau_{k_1,\ldots,k_d}^j \sigma\left(\frac{x_j - \chi_{k_j}^j}{\gamma_S}\right) \right\}$ satisfies the sum-to-zero condition, similarly to Step 2 for $|S| = 1$, it is expressed as

$$\prod_{j=1}^{d} \left\{ \eta_{k_1,\ldots,k_d}^j \left(1 - \sigma\left(\frac{x_j - \chi_{k_j}^j}{\gamma_S}\right)\right) + \tau_{k_1,\ldots,k_d}^j \sigma\left(\frac{x_j - \chi_{k_j}^j}{\gamma_S}\right) \right\} = \beta_{S,k_1,\ldots,k_d} \phi_S(\mathbf{x}_S|\theta_{S,k_1,\ldots,k_d})$$

for some $\beta_{S,k_1,\ldots,k_d}$ and $\theta_{S,k_1,\ldots,k_d}$, which implies that $f_{\mathcal{E},S}(\cdot)$ is decomposed into the sum of $(K-1)^d$ TPNNs, i.e., we have

$$f_{\mathcal{E},S}(\mathbf{x}_S) = \sum_{k=1}^{(K-1)^d} \beta_{S,k} \phi_S(\mathbf{x}_S|\theta_{S,k}).$$

$\square$

# B. Details of the experiments

All experiments are run with RTX 3090, RTX 4090, and 24GB memory.

## B.1. Details for Synthetic datasets

Table 11: **Test suite of synthetic functions.**

| $f^{(1)}$ | $Y = 10X_1 + 10X_2 + 20(X_3 - 0.3)(X_3 - 0.6) + 20X_4 + 5X_5 + 10\sin(\pi X_1 X_2) + \epsilon$ |
|---|---|
| $f^{(2)}$ | $Y = \pi^{X_1 X_2}\sqrt{2X_3} - \sin^{-1}(X_4) + \log(X_3 + X_5) - \frac{X_9}{X_{10}}\sqrt{\frac{X_7}{X_8}} - X_2 X_7 + \epsilon$ |
| $f^{(3)}$ | $Y = \exp|X_1 - X_2| + |X_2 X_3| - X_3^{2|X_4|} + \log(X_4^2 + X_5^2 + X_7^2 + X_8^2) + X_9 + \frac{1}{1+X_{10}^2} + \epsilon$ |

Table 12: **Distributions of input features corresponding to each synthetic function.**

| $f^{(1)}$ | $X_1, X_2, X_3, X_4, X_5 \sim^{iid} U(0,1)$ |
|---|---|
| $f^{(2)}$ | $X_1, X_2, X_3, X_6, X_7, X_9 \sim^{iid} U(0,1)$ and $X_4, X_5, X_8, X_{10} \sim^{iid} U(0.6, 1)$. |
| $f^{(3)}$ | $X_1, X_2, X_3, X_4, X_5, X_6, X_7, X_8, X_9, X_{10} \sim^{iid} U(-1, 1)$ |

The synthetic function $f^{(1)}$ is a slightly modified version of Friedman's synthetic function used in (Chipman et al., 2010). $f^{(2)}$ and $f^{(3)}$ are taken from the interaction detection experiments in (Tsang et al., 2017). We generate 15K data samples from the distribution in the Table 12 and functions in the Table 11. Also, we divide them into train, validation and test datasets with ratio 0.7, 0.1 and 0.2, respectively. For all of the synthetic functions, the number of basis neural networks for each component $S \subseteq [p]$, $K_S$, is set to be 30.

## B.2. Details of the experiments with real datasets.

Table 13: **Descriptions of real datasets.**

| Dataset | Size | Number of features | Problem | Number of Class |
|---|---|---|---|---|
| CALHOUSING | 21k | 8 | Regression | - |
| WINE | 4k | 11 | Regression | - |
| ONLINE | 40k | 58 | Regression | - |
| ABALONE | 4k | 10 | Regression | - |
| FICO | 10k | 23 | Classification | 2 |
| CHURN | 7k | 39 | Classification | 2 |
| CREDIT | 284k | 30 | Classification | 2 |
| LETTER | 20k | 16 | Classification | 2 |
| DRYBEAN | 13k | 16 | Classification | 7 |
| MICROSOFT | 960k | 136 | Regression | - |
| YAHOO | 700k | 699 | Regression | - |
| MADELON | 2.6k | 500 | Classification | 2 |
| CELEBA | 200K | | Classification | 2 |

**Implementation of baseline model.** We conduct experiments for all baseline models (NAM, NBM, NODE-GAM) using the official source codes. For DNN, we utilize the pytorch python package (Paszke et al., 2019) and for XGB, we utilize the xgboost package (Chen & Guestrin, 2016).

**Data descriptions.** Table 13 summarizes the descriptions of 9 real datasets analyzed in the numerical studies.

**Data preprocessing.** Minimax scaling is applied to NAM and NBM, while the standardization is used for NODE-GAM, DNN, and XGB. For ANOVA-TPNN, we transform each input feature based on the empirical quantiles to make the marginal empirical distribution of the transformed input features be close to the uniform distribution. Additionally, all categorical input variables are encoded using the one-hot encoding.

**Learning rate.** For all models except XGB, we set the learning rate of Adam optimizer 5e-3 and batch size 4096. We find the optimal learning rate of XGB via grid search.

**Model hyperparameters.** We set the architectures of NAM and NBM as defined in (Agarwal et al., 2021) and (Radenovic et al., 2022). In other words, in $\text{NA}^1\text{M}$, the dimensions of the hidden layers of each component consists of [64,32,16] for MICROSOFT, YAHOO and MADELON, and [64,64,32] for other datasets. In $\text{N}A^2\text{M}$, the hidden layers consist of [64,32,16] for the ONLINE, CREDIT and DRYBEAN datasets, [64,16,8] for MICROSOFT, YAHOO and MADELON, and [64,64,32] for the other datasets.

For XGB, DNN, and NODE-GAM, we randomly split the train, validation and test data into the ratio 70/10/20 and evaluated its performance on the validation dataset using the model trained on the train dataset. We repeated this process 10 times with randomly split data, resulting in 10 prediction performance values for the validation datasets. Then, we selected the optimal hyper-parameters by the grid search based on the average of the prediction performance values for the validation datasets.

Finally, with the optimal hyper-parameters selected earlier, we fixed the model's hyper-parameters and used the 10 train-test dataset pairs obtained from the previous data splitting to train the model on the train datasets and evaluate its performance on the test datasets.

For XGB, the range of hyper-parameters for the grid search is as below.

- The number of tree : $\{50,100,200,300,400,500,600,700,800,900,1000\}$

- max depth : $\{3, 5, 7\}$

- learning rate : $\{0.0001, 0.005, 0.01, 0.05, 0.1\}$

The hyper-parameters for NODE-GA$^1$M and NODE-GA$^2$M is determined through grid search, using similar settings to those employed in (Chang et al., 2021). That is, the range of hyper-parameters for the grid search is as below.

- The number of layer : $\{2, 4, 8\}$

- tree depth : $\{6, 8\}$

- The number of trees in each layer : $\{256, 512\}$

For DNN, we report the best prediction performance among the three architectures, as in (Radenovic et al., 2022).

- 2 hidden layers with 2048 and 1024 units

- 3 hidden layers with 1024, 512, and 512 units

- 5 hidden layers with 128, 128, 64, 64, and 64 units

For Spline-GAM, we implement it using pygam python package (Servén & Brummitt, 2018). We set the number of knot as 20 for each components and set $\lambda_S$ to be the same for all components $S$, i.e., $\lambda_S = \lambda$ for all $S \subseteq [p]$. Also, similar to the approach taken in (Chang et al., 2021), we find the best $\lambda$ penalty by using grid search on the range as below.

$$\text{Range of } \lambda = [0.001, 0.01, 0.1, 0.3, 0.6, 1, 10, 100, 1000]$$

Due to the limitation of our computational environment, in ANOVA-TPNN, for all real datasets except ONLINE, CREDIT, MICROSOFT, YAHOO, and MADELON, we select the optimal hyper-parameters by the grid search on the range $K_S$ : $\{10, 30, 50, 100\}$. For other datasets, we use the range $K_S$ : $\{10, 30, 50\}$ for grid search.

**Selected components by NID for high-dimensional datasets.** Table 14 presents the number of components used in training ANOVA-T$^2$PNN and baseline models. All main effects are used, and the second order interactions are selected using NID (Tsang et al., 2017). To find the optimal number of second order interactions, we conduct grid search over [100, 300, 500]. For MICROSOFT, 300 second order interactions are used; 500 for YAHOO, 500; and 300 for MADELON.

Table 14: **The number of components used in training ANOVA-T$^2$PNN, NA$^2$M, and NB$^2$M.**

| Dataset | MICROSOFT | YAHOO | MADELON |
|---|---|---|---|
| # of selected components | 136(Main) + 300(2nd) | 699(Main) + 500(2nd) | 500(Main) + 300(2nd) |

### B.3. Experiment details for image dataset.

For CELEBA image dataset, we use the Concept Bottleneck Model (CBM) in (Koh et al., 2020). The main idea of CBM(Koh et al., 2020) is not to directly input the embedding vector derived from image data through a CNN into a classifier for classification. Instead, CNN predicts given concepts (attributes) for the image, and the predicted values for these concepts are then used as an input of the final classifier. (Koh et al., 2020) used DNN for the final classifier which is a black box model. In this paper, we replace DNN with ANOVA-TPNN, NAM, NBM and NODE-GAM. For CNN that predicts concepts, we use linear heads for each concept on the top of the pretrained ResNet18.

All models are trained via the Adam optimizer with the learning rate 1e-3 and the batch size for training 256. For ANOVA-T$^1$PNN, $K_S$ is determined through grid search on [10,30,50]. For NA$^1$M and NB$^1$M, the neural network consists of 3 hidden layer with sizes (64,64,32) and (256,128,128), respectively. Due to the limitations of the computational environment, for ANOVA-T$^2$PNN, we set $K_S = 10$ for the main effects and $K_S = 3$ for the second order interactions. For NA$^2$M, we use the neural network consisting of 3 hidden layers with sizes (16,16,8), and for NB$^2$M, the neural network consists of 3 hidden layers with sizes (128,64,64). For NODE-GA$^1$M and NODE-GA$^2$M, the number of trees and the number of layers are determined through grid search on the range as :

- The number of layers : $\{2, 4\}$

- Tree depth : $\{4, 6\}$

- The number of trees in each layer : $\{50, 125, 256\}$

### B.4. Experiment details for component selection

An important implication of stable estimation of the components is the ability of selecting signal components. That is, ANOVA-TPNN can effectively identify signal components in the true function by measuring the variations of the estimated components. For example, we consider the $l_1$ norm of each estimated component (i.e, $\|f_S(\mathbf{x}_S)\|_1$) as the important score, and select the components whose important scores are large. This simple component selection procedure would not perform well if component estimation is unstable.

To investigate how well ANOVA-TPNN selects the true signal components, we conduct an experiment similar to the one in (Tsang et al., 2017). We generate synthetic datasets from $Y = f(\mathbf{x}) + \epsilon$, where $f$ is the true prediction model and $\epsilon$ is a noise generated from a Gaussian distribution with mean 0 and variance $\sigma_\epsilon^2$. Then, we apply ANOVA-T$^2$PNN, NA$^2$M and NB$^2$M to calculate the importance scores of the main effects and second order interactions and examine how well they predict whether a given component is signal. For the performance measure of component selection, we use AUROC obtained from the pairs of $\|\hat{f}_S\|_1$ and $r_S$ for all $S \subset [p]$ with $|S| \leq 2$, where $\hat{f}_S$ are the estimates of $f_S$ in $f$ and $r_S = \mathbb{I}(\|f_S^{(k)}\|_1 > 0)$ are the indicators whether $f_S$ are signal or not.

For the true prediction model, we consider the three functions $f^{(k)}, k = 1, 2, 3$ whose details are given in Appendix B.1. We set the data size to 15,000 and set $\sigma_\epsilon^2$ to make the signal-to-noise ratio become 5.

## C. Ablation studies.

### C.1. The choice of $K_S$ in ANOVA-TPNN.

Table 15 presents the averages (the standard deviations) of prediction performance of ANOVA-TPNN on 10 randomly sampled datasets from ABALONE for various values of $K_S$. For simplicity, we set all $K_S, S \subseteq [p]$ to be equal. We observe that $K_S$ around 50 yields the best results for ANOVA-T$^1$PNN and $K_S$ around 10 for ANOVA-T$^2$PNN. The results suggest that the choice of optimal $K_S$ is important for prediction performance and a smaller $K_S$ is required for a model with higher order interactions.

Table 15: **Results of prediction performance for various $K_S$ on ABALONE dataset.**

| $K_S$ | 1 | 5 | 10 | 50 | 100 |
|---|---|---|---|---|---|
| ANOVA-T$^1$PNN | 2.176 (0.09) | 2.163 (0.08) | 2.160 (0.09) | 2.135 (0.09) | 2.159 (0.08) |
| ANOVA-T$^2$PNN | 2.103 (0.08) | 2.102 (0.08) | 2.087 (0.08) | 2.105 (0.08) | 2.122 (0.08) |

### C.2. Impact of the initial parameter values to stability

We investigate how the choice of initial parameter values affects the stability of the estimated components of ANOVA-TPNN by analyzing synthetic datasets generated from $f^{(1)}$. We fit ANOVA-T$^2$PNN on 5 independently generated datasets, and the 5 estimated main effects are presented in Figure 3. We observe ANOVA-T$^2$PNN is insensitive to the choice of initial parameter values.

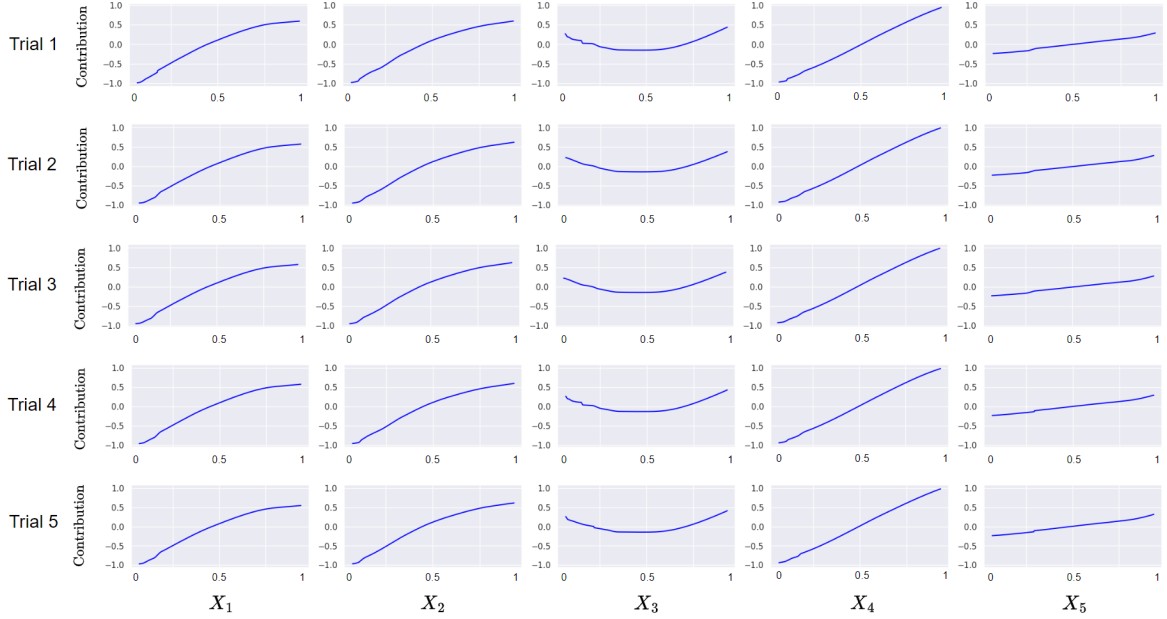

Figure 3: **Plots of the functional relations of the main effects in ANOVA-T$^2$PNN on synthetic datasets generated from $f^{(1)}$.**

## D. Prediction performance and Interpretability on CELEBA dataset.

**Comparison with baseline models in terms of prediction performance.** We consider two types of CBMs: one is the joint concept bottleneck model (JCBM), where the CNN and the classifier are trained simultaneously, and the other is the independent concept bottleneck model (ICBM), where the CNN is kept fixed, and only the classifier is trained. In Table 16, we compare prediction performance of various models for the final classifier in the JCBM, which shows the prediction performances of ANOVA-T$^1$PNN and ANOVA-T$^2$PNN are comparable or superior to their competitors. In Table 17, we compare the prediction performance of ANOVA-T$^1$PNN, NA$^1$M, NB$^1$M, DNN, and Linear model. The hidden layers of DNN are consists of five layers with size of (128,128,64,64,64). In ICBM, the prediction performance is inferior to other nonlinear models when the classifier is linear. However, ANOVA-T$^1$PNN is found to have comparable prediction performance compared to other baseline models including NA$^1$M, NB$^1$M, and DNN.

**Prediction performance with and without the monotone constraint.** Table 18 presents the prediction performances of ANOVA-TPNN with and without the monotone constraint. For attributes 'Bald', 'Big Nose', 'Goatee' and 'Mustache', we apply the increasing monotone constraint, while for attributes 'Arched Eyebrows', 'Attractive', 'Heavy Makeup', 'No Beard', 'Wearing Earrings', 'Wearing Lipstick', 'Wearing Necklace', 'Wearing Necktie', we use the decreasing monotone constraint. Prediction performances are similar regardless of the monotone constraint but interpretation of the estimated model can be quite different which is discussed in the followings.

**Global interpretation on CELEBA dataset.** In ANOVA-T$^1$PNN without the monotone condition, we select the two features which have incorrect global interpretations among the 10 most important ones. Table 19 gives the importance scores (normalized by of the maximum important score) and its ranks of 2 components obtained by ANOVA-T$^1$PNN on a randomly sampled data from CELEBA dataset.

**Local interpretation on CELEBA dataset.** In Table 20, we observe that Image 2-1 of Figure 4 is correctly classified when the monotone constraint is applied, whereas it is misclassified without the monotone constraint. Despite Image 2-1 of Figure 4 having 'No Beard', 'Heavy Makeup', and 'Wearing Lipstick', the scores of these features without the monotone constraint make the probability of being male increase. However, ANOVA-T$^1$PNN with the monotone constraint can avoid these unreasonable interpretations and classifies the image correctly.

In Image 2-2 of Figure 4, we observe that ANOVA-T$^1$PNN with the monotone constraint assigns a negative score to 'No Beard' that increases the probability of being classified as female. However, ANOVA-T$^1$PNN without the monotone condition assigns a positive score to 'No Beard' that increases the probability of being classified as male, even though 'No Beard' is present.

Note that we can understand why ANOVA-T$^1$PNN with the monotone constraint classifies Image 2-2 of Figure 4 incorrectly because there is no bear in the image. In contrast, it is not easy to understand why ANOVA-T$^1$PNN without the monotone constraint classifies Image 2-1 of Figure 4 incorrectly. That is, imposing the monotone constraint is helpful to learn more reasonably interpretable models.

**Attributes to which monotone constraints are applied.** For attributes 'Bald', 'Big Nose', 'Goatee' and 'Mustache', we apply the increasing monotone constraint, while for attributes 'Arched Eyebrows', 'Attractive', 'Heavy Makeup', 'No Beard', 'Wearing Earrings', 'Wearing Lipstick', 'Wearing Necklace', 'Wearing Necktie', we used the decreasing monotone constraint.

Table 16: **Accuracies (standard deviations) on CELEBA dataset in JCBM.**

| ANOVA-T$^1$PNN | NODE-GA$^1$M | NA$^1$M | NB$^1$M | ANOVA-T$^2$PNN | NODE-GA$^2$M | NA$^2$M | NB$^2$M |
|---|---|---|---|---|---|---|---|
| 0.985 (0.001) | 0.981 (0.006) | 0.982 (0.002) | 0.980 (0.002) | **0.986** (0.001) | 0.981 (0.006) | **0.986** (0.001) | 0.980 (0.002) |

Table 17: **Accuracies (standard deviations) on CELEBA dataset in ICBM.**

| ANOVA-T$^1$PNN | NA$^1$M | NB$^1$M | DNN | Linear |
|---|---|---|---|---|
| 0.929 (0.001) | 0.927 (0.001) | 0.930 (0.001) | 0.936 (0.001) | 0.876 (0.006) |

Table 18: **Results of prediction performance of ANOVA-TPNN with and without the monotone constraint.**

| | Measure | ANOVA-T$^1$PNN | ANOVA-T$^2$PNN |
|---|---|---|---|
| Without Monotone constraint | Accuracy ↑ | 0.985 (0.001) | 0.986 (0.001) |
| With Monotone constraint | Accuracy ↑ | 0.984 (0.001) | 0.985 (0.001) |

Table 19: **Importance scores and ranks for the 3 important components.**

| Components | Monotone | No Beard | Wearing Lipstick |
|---|---|---|---|
| Score (Rank) | X | 0.794 (3) | 0.465 (4) |
| Score (Rank) | O | 0.757 (6) | 0.738 (7) |

Table 20: **Results of local interpretation with and without the monotone constraint.**

| Image index | Monotone | Heavy Makeup | No beard | Wearing Lipstick | classified label | True label |
|---|---|---|---|---|---|---|
| 2-1 | X | 0.030 | 0.035 | 0.093 | male | female |
| 2-1 | O | -0.080 | -0.161 | -0.106 | female | female |
| 2-2 | X | 0.036 | 0.104 | 0.095 | male | male |
| 2-2 | O | -0.081 | -0.183 | -0.106 | female | male |

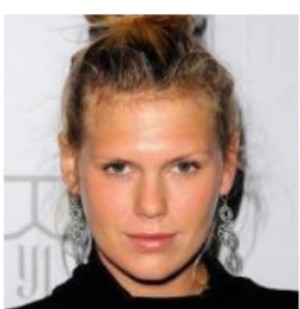 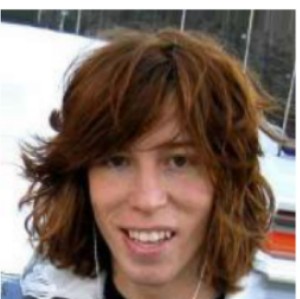

Figure 4: **Two misclassified images.**

# E. Additional experiments for the stability of ANOVA-TPNN.

## E.1. Stability of the estimated components with respect to variations of training data

We investigate the stability of components estimated by ANOVA-TPNN when training datasets vary. We analyze CALHOUSING (Pedregosa et al., 2011a) and WINE (Cortez et al., 2009) datasets and compare ANOVA-TPNN with NAM and NBM in view of stability of component estimation. For 5 randomly sampled training and test datasets, we train ANOVA-TPNN, NAM, and NBM on the training datasets and plot the functional relations of the main effects on the test datasets.

**Experiment for CALHOUSING dataset.** Figures 5, 6 and 7 present the plots of the functional relations of the main effects estimated by ANOVA-T$^1$PNN, NA$^1$M, and NB$^1$M for 5 randomly sampled training datasets, and Figures 8, 9 and 10 present the plots of the functional relations of the main effects estimated by ANOVA-T$^2$PNN, NA$^2$M, and NB$^2$M for 5 randomly sampled datasets. We observe that the 5 estimates of each component estimated by ANOVA-TPNN are much more stable compared to those by NAM and NBM. Note that in Figure 9, we observe that some components are estimated as a constant function in NA$^2$M, which would be partly because the main effects are absorbed into the second order interactions.

**Experiment for WINE dataset.** Figures 11, 12 and 13 present the plots of the functional relations of the main effects estimated by ANOVA-T$^1$PNN, NA$^1$M, and NB$^1$M for 5 randomly sampled datasets, and Figures 14, 15 and 16 present the plots of the functional relations of the main effects estimated by ANOVA-T$^2$PNN, NA$^2$M, and NB$^2$M for 5 randomly sampled datasets. The results are similar to those of CALHOUSING dataset.

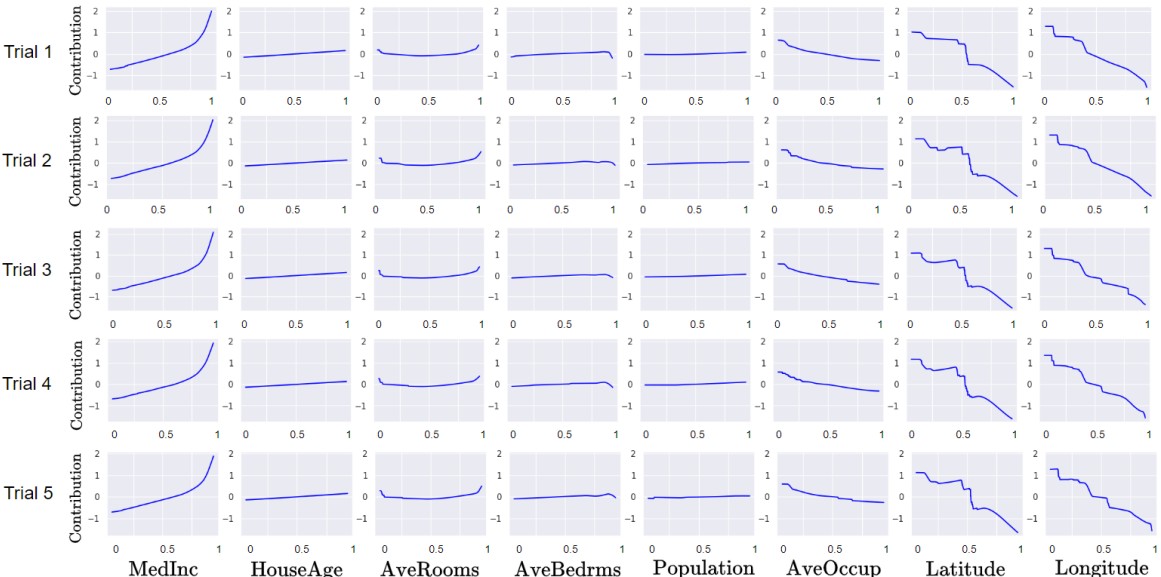

Figure 5: **Plots of the functional relations of the main effects in ANOVA-T$^1$PNN on CALHOUSING dataset.**

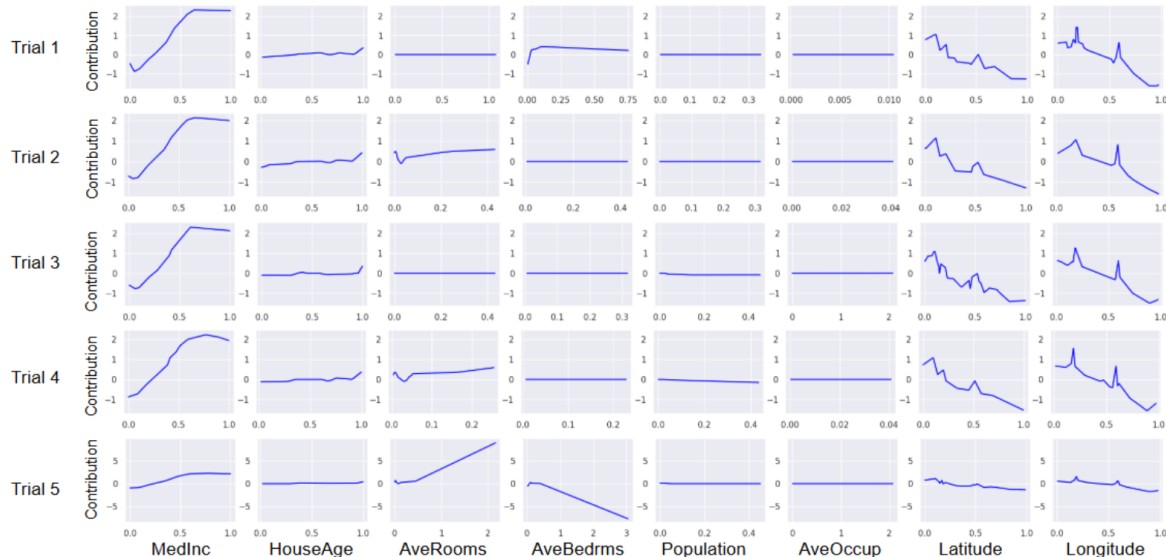

Figure 6: **Plots of the functional relations of the main effects in NA$^1$M on CALHOUSING dataset.**

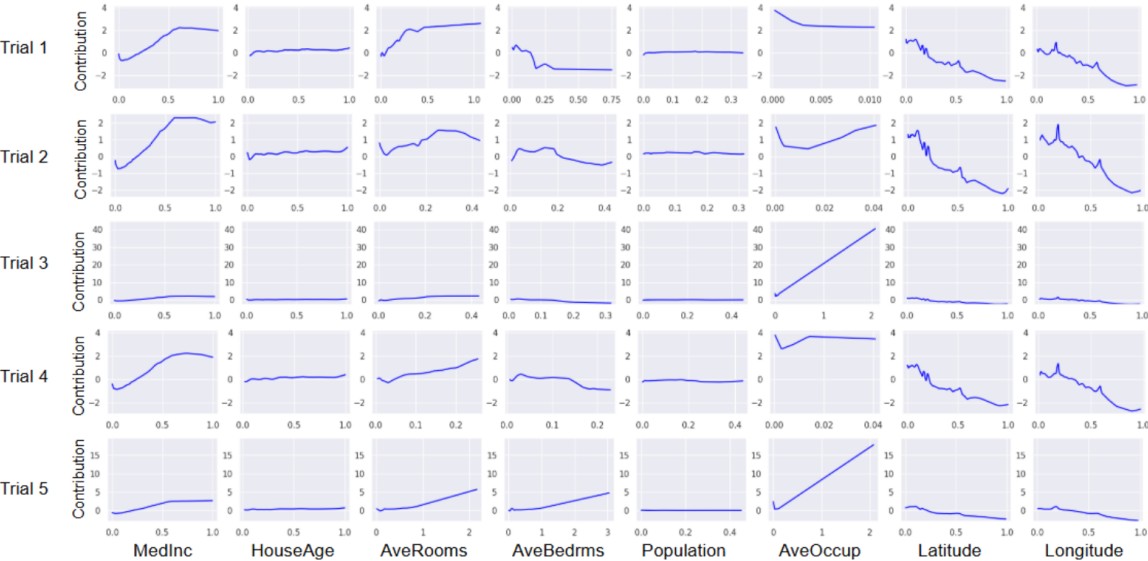

Figure 7: **Plots of the functional relations of the main effects in NB$^1$M on CALHOUSING dataset.**

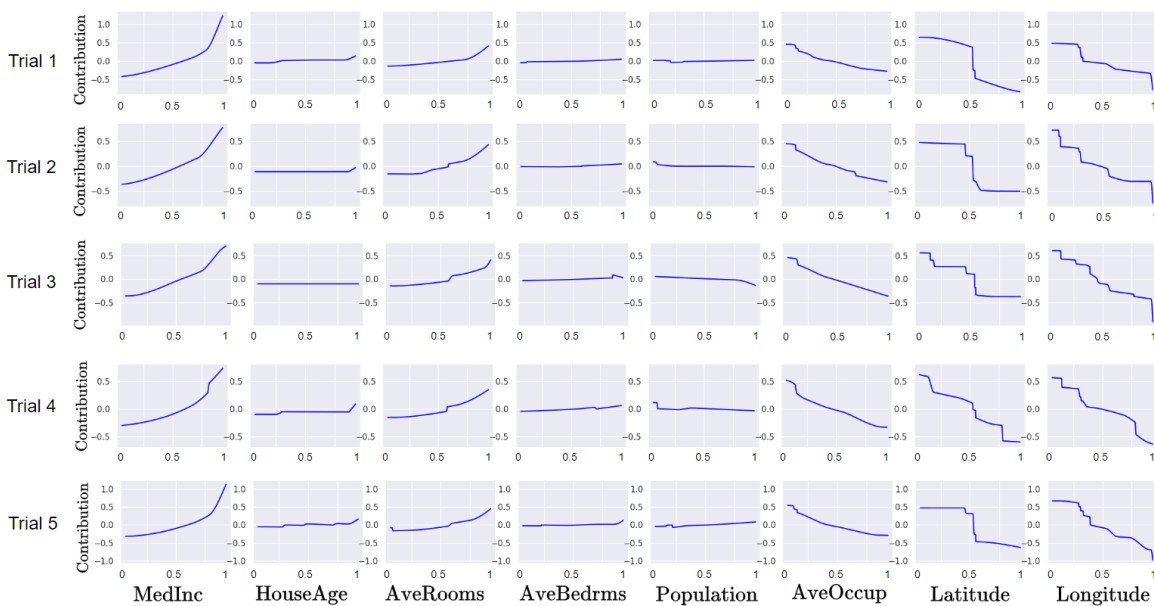

Figure 8: **Plots of the functional relations of the main effects in ANOVA-T$^2$PNN on CALHOUSING dataset.**

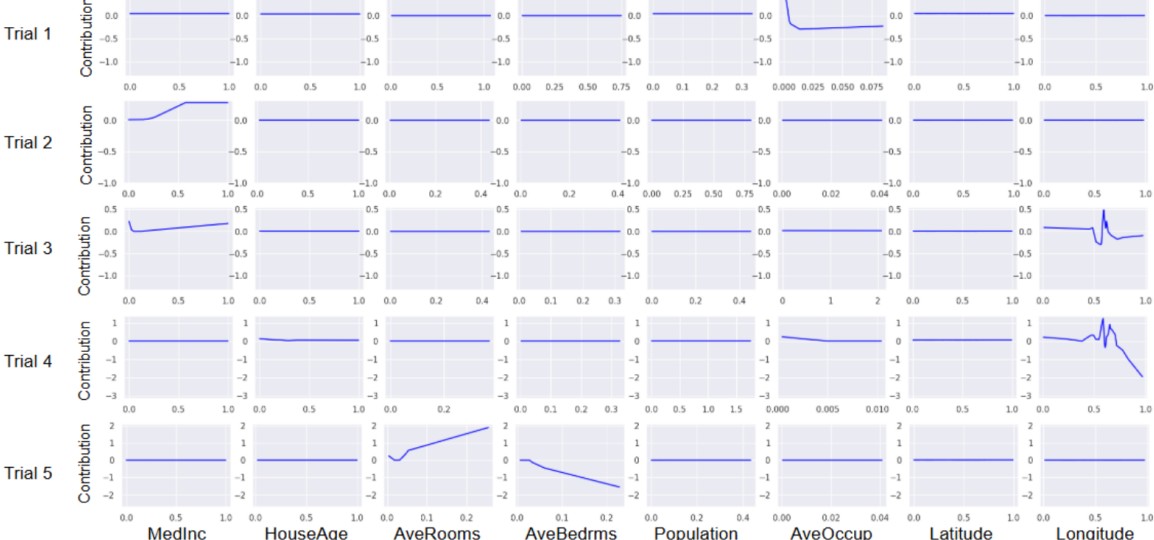

Figure 9: **Plots of the functional relations of the main effects in NA$^2$M on CALHOUSING dataset.**

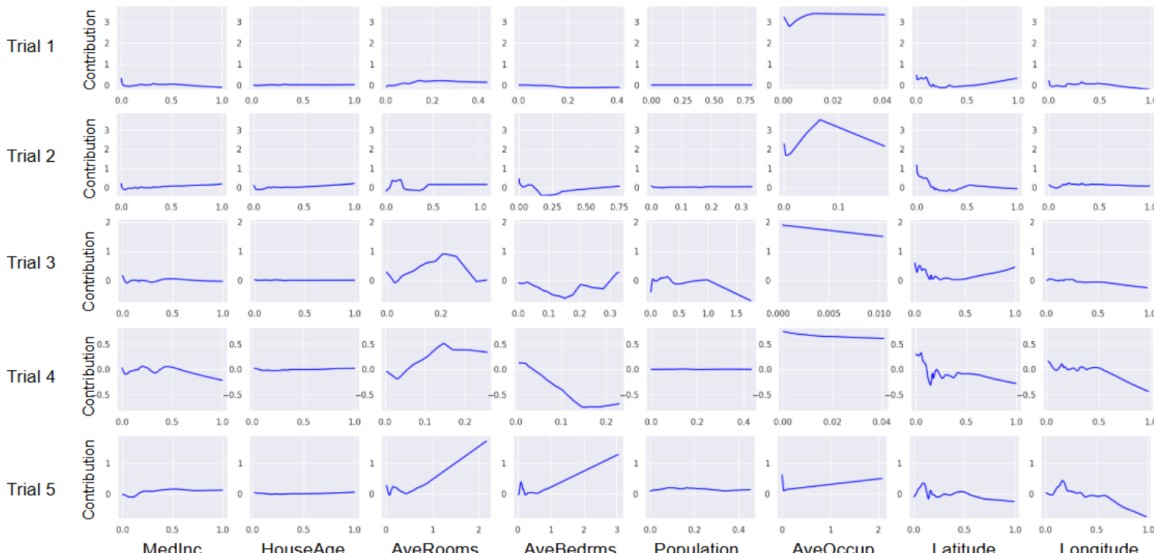

Figure 10: **Plots of the functional relations of the main effects in NB$^2$M on CALHOUSING dataset.**

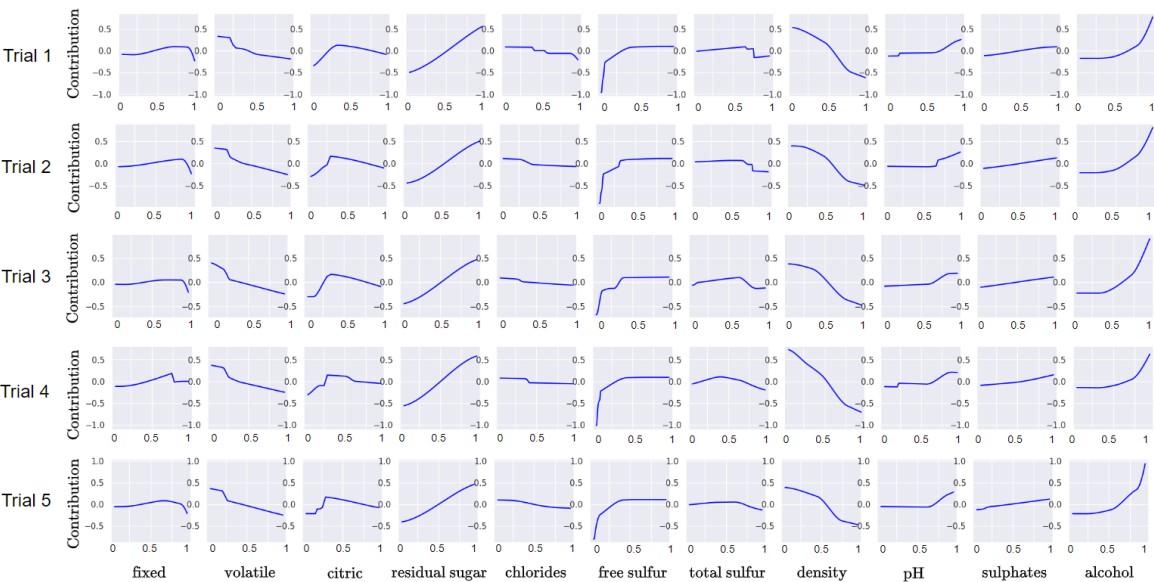

Figure 11: **Plots of the functional relations of the main effects in ANOVA-T$^1$PNN on WINE dataset.**

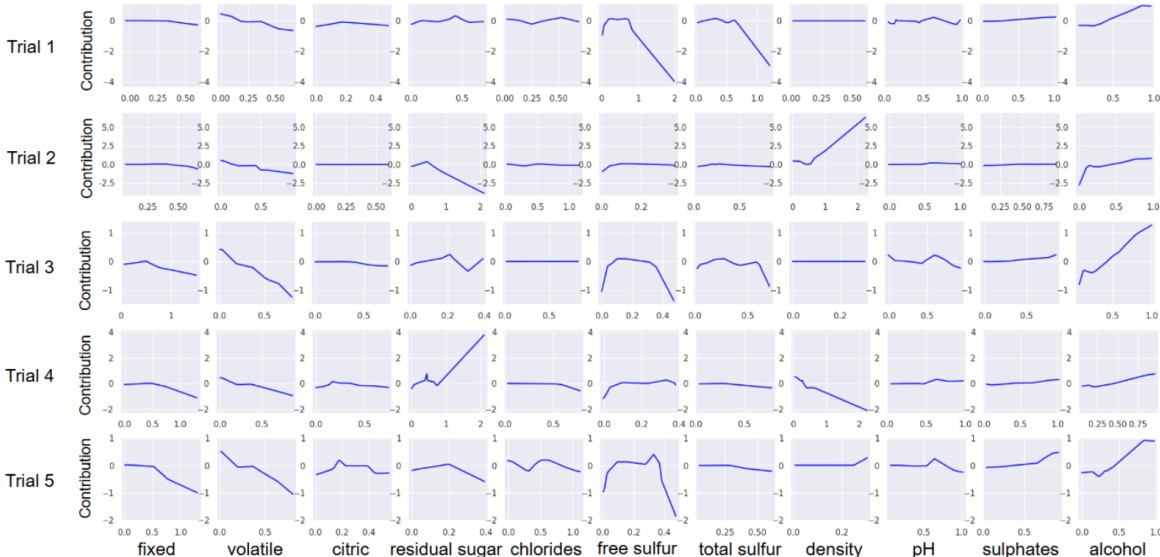

Figure 12: **Plots of the functional relations of the main effects in NA$^1$M on WINE dataset**.

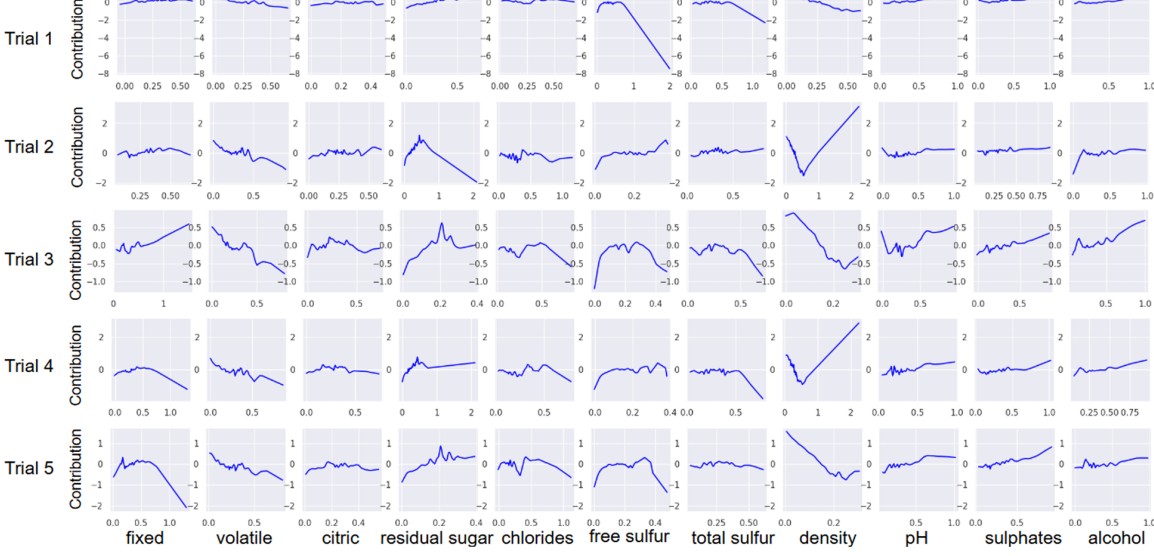

Figure 13: **Plots of the functional relations of the main effects in NB$^1$M on WINE dataset**.

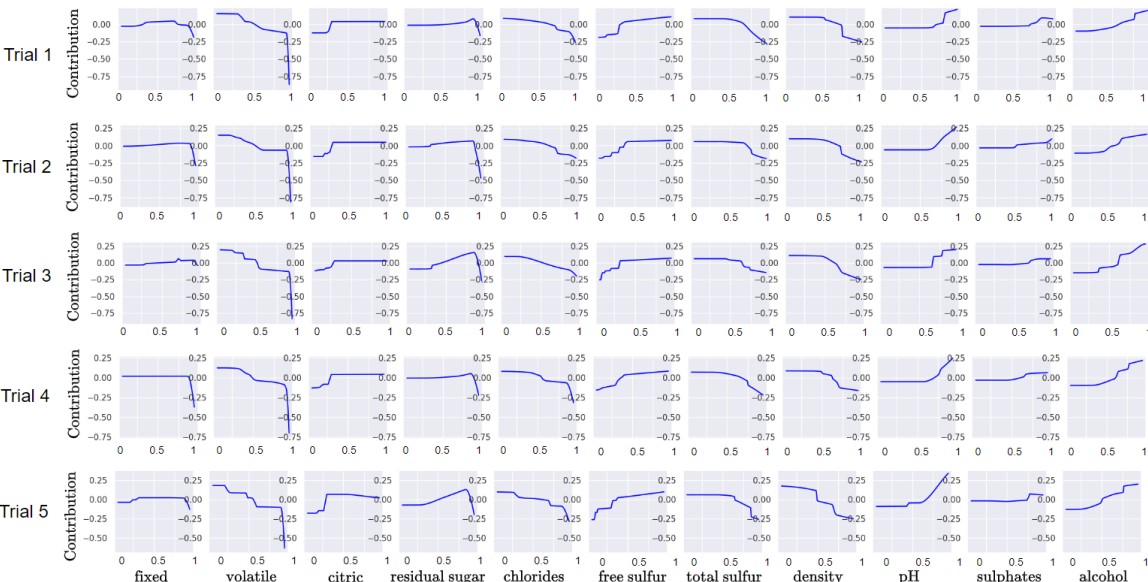

Figure 14: **Plots of the functional relations of the main effects in ANOVA-T$^2$PNN on WINE dataset**.

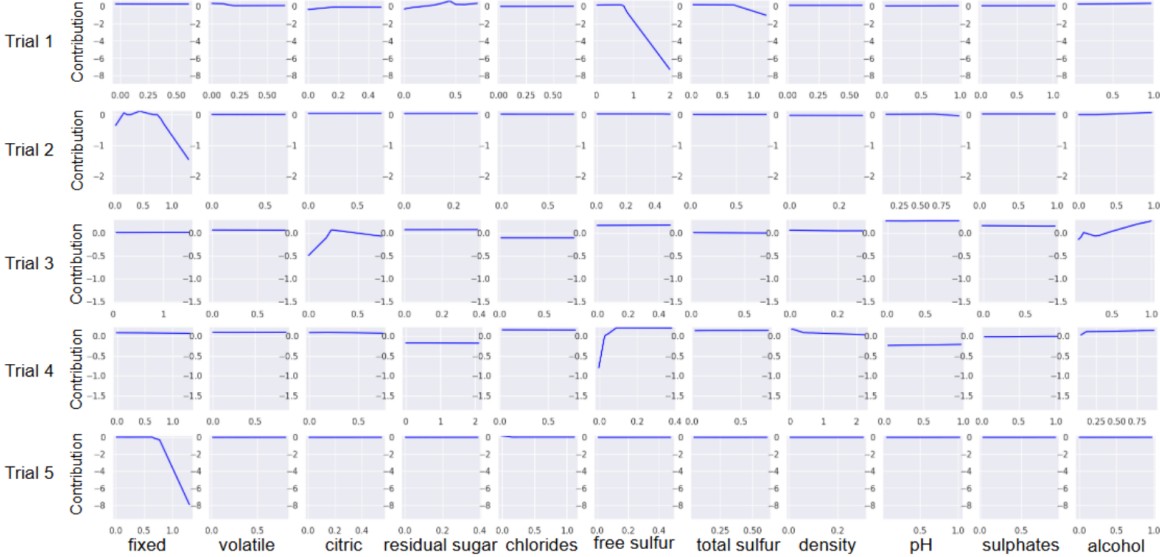

Figure 15: **Plots of the functional relations of the main effects in NA$^2$M on WINE dataset**.

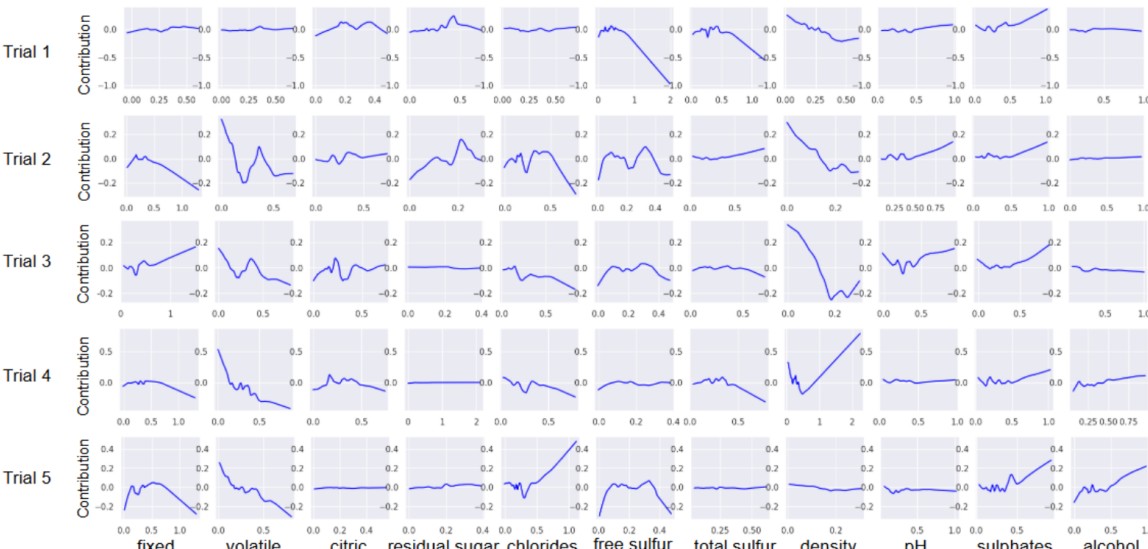

Figure 16: **Plots of the functional relations of the main effects in NB$^2$M on WINE dataset**.

# F. Stability of NBM-TPNN.

Figures 17 and 18 show the plots of the functional relations of the main effects estimated by NBM-T[1]PNN on 5 randomly sampled datasets from WINE and CALHOUSING dataset, respectively, which amply show that NBM-TPNN is also highly stable in estimation of components.

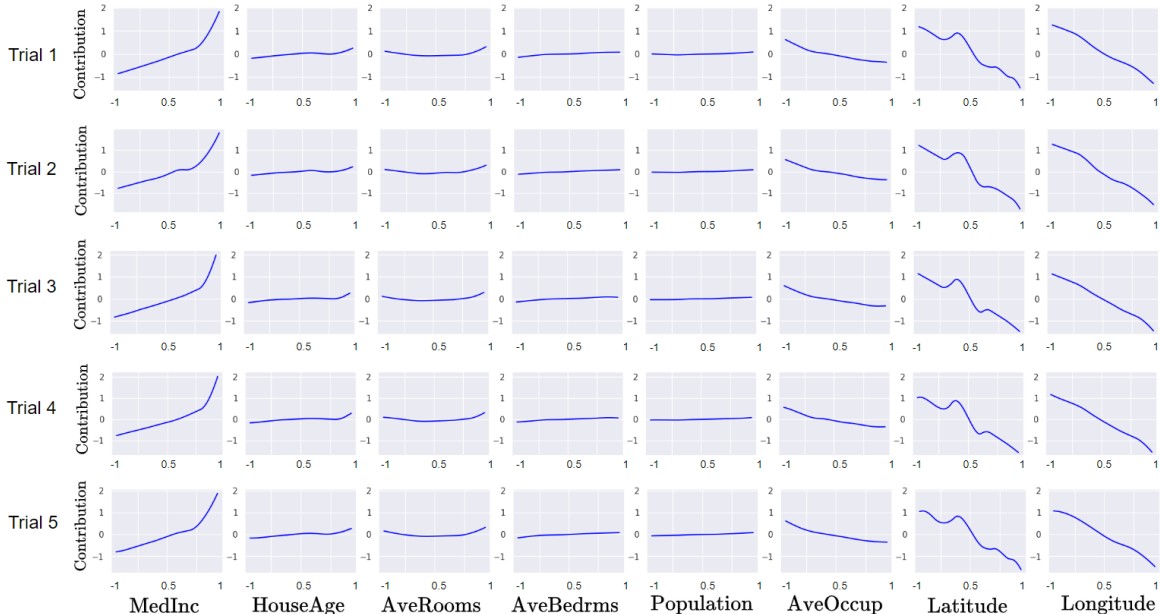

Figure 17: **Plots of the functional relations of the main effect estimated by NBM-T[1]PNN on 5 randomly sampled training data from CALHOUSING dataset.**

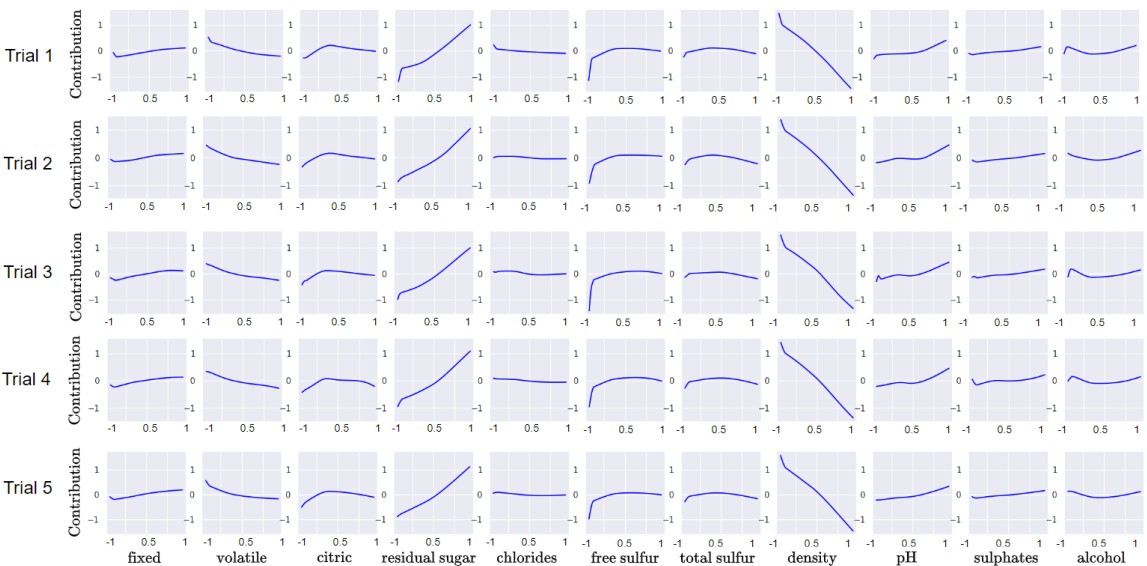

Figure 18: **Plots of the functional relations of the main effects estimated by NBM-T[1]PNN on 5 randomly sampled training data from WINE dataset.**

# G. Additional experiments for component selection

Table 21 presents the averages (the standard deviations) of the prediction performance of the models used in the component selection experiment in Section 4.2, which indicates that ANOVA-TPNN, NAM and NBM perform similarly.

Table 21: **The results of prediction performance.** We report the averages and standard deviations of RMSEs of ANOVA-T$^2$PNN, NA$^2$M and NB$^2$M on 10 synthetic datasets generated from $f^{(1)}$, $f^{(4)}$ and $f^{(3)}$.

| | | GA$^2$M | | |
|---|---|---|---|---|
| Synthetic function | Measure | ANOVA T$^2$PNN | NA$^2$M | NB$^2$M |
| $f^{(1)}$ | RMSE $\downarrow$ | 3.483 (0.03) | **3.474** (0.03) | 3.511 (0.03) |
| $f^{(2)}$ | RMSE $\downarrow$ | 0.076 (0.001) | 0.088 (0.005) | **0.075** (0.001) |
| $f^{(3)}$ | RMSE $\downarrow$ | 0.161 (0.003) | 0.183 (0.016) | **0.137** (0.003) |

## H. Comparison of ANOVA-SHAP and Kernel-SHAP

In this section, we conduct an experiment to investigate the similarity between ANOVA-SHAP and Kernel-SHAP (Lundberg, 2017). We calculate ANOVA-SHAP from ANOVA-$T^1$PNN approximating a given black-box model. For the black-box model, we use XGB with 100 trees, a maximum depth of 7, and a learning rate of 0.1 trained on WINE dataset. Below, we compare the SHPA values for three inputs.

For input $\mathbf{x}_1 = (6.9, 0.22, 0.32, 9.3, 0.04, 22, 110, 0.99, 3.34, 0.54, 10.7)^\top$, the SHAP values are given as:

$$\text{Kernel-SHAP} : (0.057, 0.095, 0.065, 0.088, -0.009, -0.136, 0.034, -0.119, 0.149, 0.040, -0.082)$$
$$\text{ANOVA-SHAP} : (-0.017, 0.056, 0.104, 0.132, 0.006, -0.118, 0.055, -0.163, 0.093, 0.016, -0.081).$$

For input $\mathbf{x}_2 = (7.00, 0.17, 0.33, 4.00, 0.03, 17, 127, 0.99, 3.19, 0.39, 10.6)^\top$, the SHAP values are given as:

$$\text{Kernel-SHAP} : (0.015, 0.327, 0.031, -0.080, 0.028, -0.146, 0.050, 0.043, -0.050, -0.01, -0.115)$$
$$\text{ANOVA-SHAP} : (-0.015, 0.243, 0.093, -0.148, 0.024, -0.111, 0.075, 0.141, -0.061, -0.022, -0.101).$$

For input $\mathbf{x}_3 = (6.90, 0.25, 0.35, 9.20, 0.03, 42, 150, 0.99, 3.21, 0.36, 11.5)^\top$, the SHAP values are given as:

$$\text{Kernel-SHAP} : (-0.004, 0.018, 0.060, 0.092, 0.032, 0.047, 0.014, -0.024, -0.123, -0.027, 0.084)$$
$$\text{ANOVA-SHAP} : (-0.017, 0.003, 0.071, 0.128, 0.024, 0.088, 0.062, -0.004, -0.062, -0.046, 0.118).$$

The results suggest that ANOVA-SHAP and Kernel SHAP are similar. At least, the signs are exactly the same. An obvious advantage of ANOVA-SHAP is computation. Only one model fitting for finding a ANOVA-TPNN approximating a given black-box model is required for ANOVA-SHAP. In contrast, Kernel SHAP requires training a linear models for each data point. For illustration, if we want to calculate the SHAP values for 1000 data points, computation time of ANOVA-SHAP is approximately 6,500 times faster than that of Kernel SHAP.

# I. Additional Experiments for prediction performance comparison with Decision Tree

Table 22: **Results of the prediction performance in Decision Tree and ANOVA-TPNN.**

| Dataset | Measure | Decision Tree | ANOVA-T$^1$PNN | ANOVA-T$^2$PNN |
|---|---|---|---|---|
| CALHOUSING | RMSE $\downarrow$ | 0.671 ( 0.02 ) | 0.614 ( 0.01 ) | 0.512 ( 0.01 ) |
| WINE | RMSE $\downarrow$ | 0.811 ( 0.03 ) | 0.725 ( 0.02 ) | 0.704 ( 0.02 ) |
| ONLINE | RMSE $\downarrow$ | 1.119 ( 0.26 ) | 1.111 ( 0.25 ) | 1.111 ( 0.25 ) |
| ABALONE | RMSE $\downarrow$ | 2.396 ( 0.08 ) | 2.135 ( 0.09 ) | 2.087 ( 0.08 ) |
| FICO | AUROC $\uparrow$ | 0.704 ( 0.02 ) | 0.799 ( 0.007 ) | 0.800 ( 0.007 ) |
| CHURN | AUROC $\uparrow$ | 0.676 ( 0.03 ) | 0.839 ( 0.012 ) | 0.842 ( 0.012 ) |
| CREDIT | AUROC $\uparrow$ | 0.890 ( 0.02 ) | 0.983 ( 0.005 ) | 0.984 ( 0.006 ) |
| LETTER | AUROC $\uparrow$ | 0.745 ( 0.001 ) | 0.900 ( 0.003 ) | 0.984 ( 0.001 ) |
| DRYBEAN | AUROC $\uparrow$ | 0.975 ( 0.0002 ) | 0.995 ( 0.001 ) | 0.997 ( 0.001 ) |

Table 22 presents the averages and standard deviations of the prediction performance of Decision Tree (Breiman, 2017) for 10 trials. We observe that the performance of ANOVA-TPNN is significantly better than that of Decision Tree. We implement Decision Tree by using the scikit-learn python package (Pedregosa et al., 2011b) and turned by using the optuna python package based on below range of hyper-parameters.

- Range of max depth = [2 ,12]

- Range of min_samples_leaf = [2,10]

- Range of min_samples_split = [2,10]

- Range of max_leaf_nodes = [2,10]

## J. Additional Experiments for runtime on various datasets.

Table 23: **Results of runtimes of NA$^1$M, NB$^1$M, ANOVA-T$^1$PNN, and NBM-T$^1$PNN.**

| Dataset | Size of dataset | # of features | NA$^1$M | NB$^1$M | ANOVA-T$^1$PNN | NBM- T$^1$PNN |
|---|---|---|---|---|---|---|
| ABALONE | 4K | 10 | 6.6 sec | 3.0 sec | 1.6 sec | 1.5 sec |
| CALHOUSING | 21K | 8 | 14.1 sec | 4.1 sec | 3.8 sec | 3.5 sec |
| ONLINE | 40K | 58 | 68 sec | 15.6 sec | 65 sec | 9.8 sec |

We conduct experiments to assess the scalability of NBM-TPNN. We consider NA$^1$M, which has 3 hidden layers with 16, 16, and 8 nodes; 10 basis DNNs for NB$^1$M, which have 3 hidden layers with 32, 16, and 16 nodes; $K_S = 10$ for each component $S$ in ANOVA-T$^1$PNN; and 10 basis functions in NBM-T$^1$PNN. Table 23 presents the results of runtimes of NA$^1$M, NB$^1$M, ANOVA-T$^1$PNN, and NBM-T$^1$PNN on ABALONE, CALHOUSING, and ONLINE datasets. When the dimension of input features is small, there is little difference in runtime between ANOVA-T$^1$PNN and NBM-T$^1$PNN. However, as the input dimension increases, the runtime gap becomes more pronounced.

## K. Additional Experiments for ReLU activation function

Table 24: **Results of prediction performance of ANOVA-TPNN with ReLU**. We report the averages of RMSE (standard deviation) and stability score for 10 trials.

|  | Dataset | ANOVA-T$^1$PNN with ReLU | ANOVA-T$^1$PNN |
|---|---|---|---|
| RMSE ↓ | ABALONE | 2.148 (0.08) | 2.135 (0.09) |
|  | WINE | 0.735 (0.02) | 0.725 (0.02) |
| Stability score ↓ | ABALONE | 0.016 | 0.008 |
|  | WINE | 0.018 | 0.011 |

We conduct additional experiments to evaluate the performance of ANOVA-TPNN with the ReLU activation function i.e., $\sigma(x) = \max(0, x)$. The $K_S$s for ANOVA-T$^1$PNNs with ReLU and sigmoid are determined through grid search on the range [10,30,50]. Table 24 presents the results of stability scores and prediction performance of ANOVA-TPNNs with the ReLU and sigmoid activations on ABALONE and WINE dataset. We observe that ANOVA-TPNN with Relu is slightly inferior to ANOVA-TPNN with sigmoid in view of both prediction performance and stability of component estimation. This would be because ANOVA-TPNN with sigmoid is more robust to input outliers than ANOVA-TPNN with ReLU.

## L. ANOVA-TPNN without sum-to-zero condition

Table 25: **Comparison of ANOVA-TPNN and GAM-TPNN**. We report the stability score (normalized by the that of ANOVA-T$^1$PNN or ANOVA-T$^2$PNN) for 10 trials.

|  | ANOVA-T$^1$PNN | GAM-T$^1$PNN | ANOVA-T$^2$PNN | GAM-T$^2$PNN |
|---|---|---|---|---|
| CALHOUSING | **1.000** | 1.500 | **1.000** | 1.690 |
| WINE | **1.000** | 2.550 | **1.000** | 1.300 |

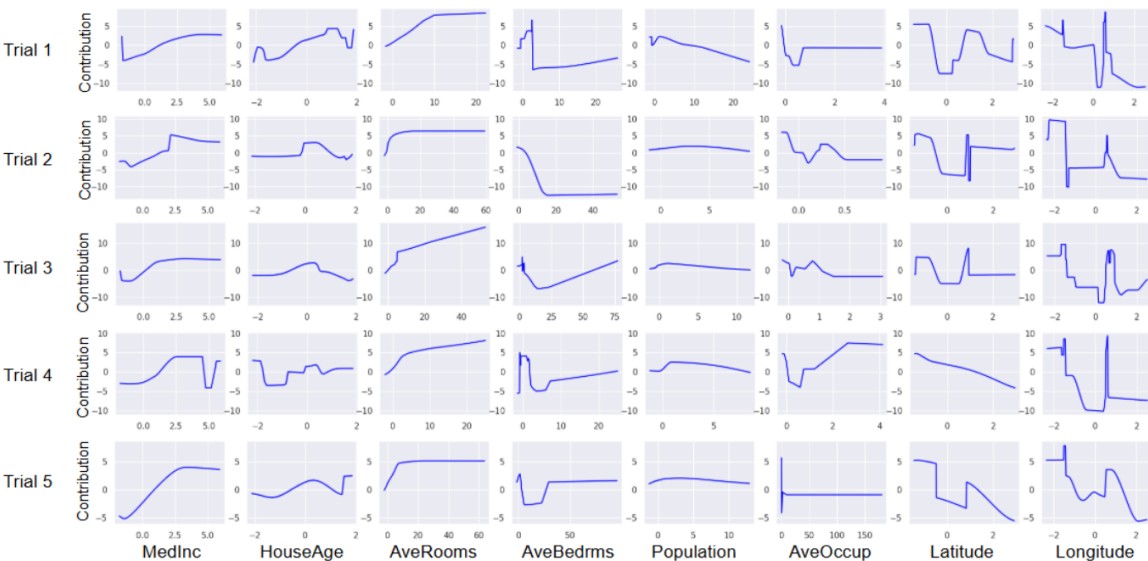

Figure 19: **Plots of the functional relations of the main effects on 5 randomly sampled training data from CALHOUSING datasets.**

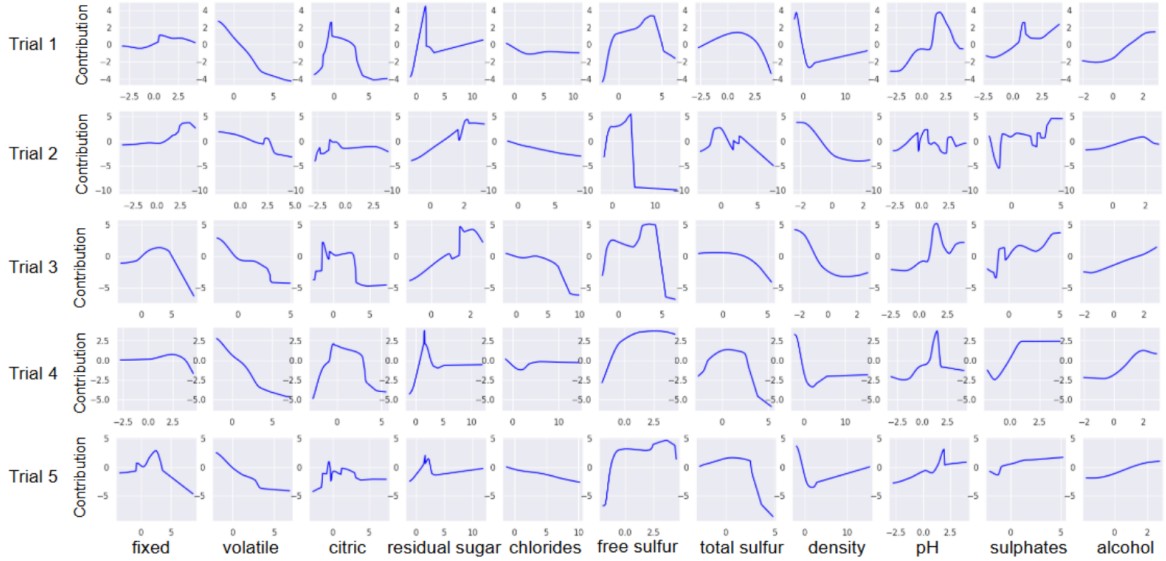

Figure 20: **Plots of the functional relations of the main effects on 5 randomly sampled training data from WINE datasets.**

We investigate the stability in component estimation of ANOVA-TPNN without the sum-to-zero condition, which we denote

GAM-TPNN, by analyzing CALHOUSING and WINE datasets. In GAM-T$^1$PNN, we approximate $f_j(x_j)$ by

$$f_j(x_j) \approx \sum_{k=1}^{K_j} \left\{ \beta_{jk}^1 \left( 1 - \sigma \left( \frac{x_j - b_{jk}}{\gamma_{jk}} \right) \right) + \beta_{jk}^2 \sigma \left( \frac{x_j - b_{jk}}{\gamma_{jk}} \right) \right\}$$

where $\beta_{jk}^1, \beta_{jk}^2, b_{jk}, \gamma_{jk}$ are learnable parameters. GAM-T$^1$PNN can be easily extended to GAM-T$^d$PNN in a similar way to ANOVA-T$^d$PNN.

Table 25 presents the stability score of ANOVA-TPNN and GAM-TPNN based on 10 randomly selected datasets. Without the sum-to-zero condition, we observe increasing in the stability score. In particular, when the second order interactions are in the model, the main effects are estimated very unstably.

Figure 19 and 20 present the plots of the functional relations of the main effects on CALHOUSING and WINE dataset in GAM-T$^2$PNN. We observe that GAM-T$^2$PNN estimates the components more unstable compared to ANOVA-T$^2$PNN.

## M. On the post-processing for the sum-to-zero condition

Table 26: **Stability scores for 'Latitude' and 'Longitude' of CALHOUSING dataset after post-processing**

| Model | ANOVA-T$^2$PNN | NA$^2$M | NB$^2$M |
|---|---|---|---|
| Latitude | 0.006 | 0.067 | 0.104 |
| Longitude | 0.015 | 0.094 | 0.103 |

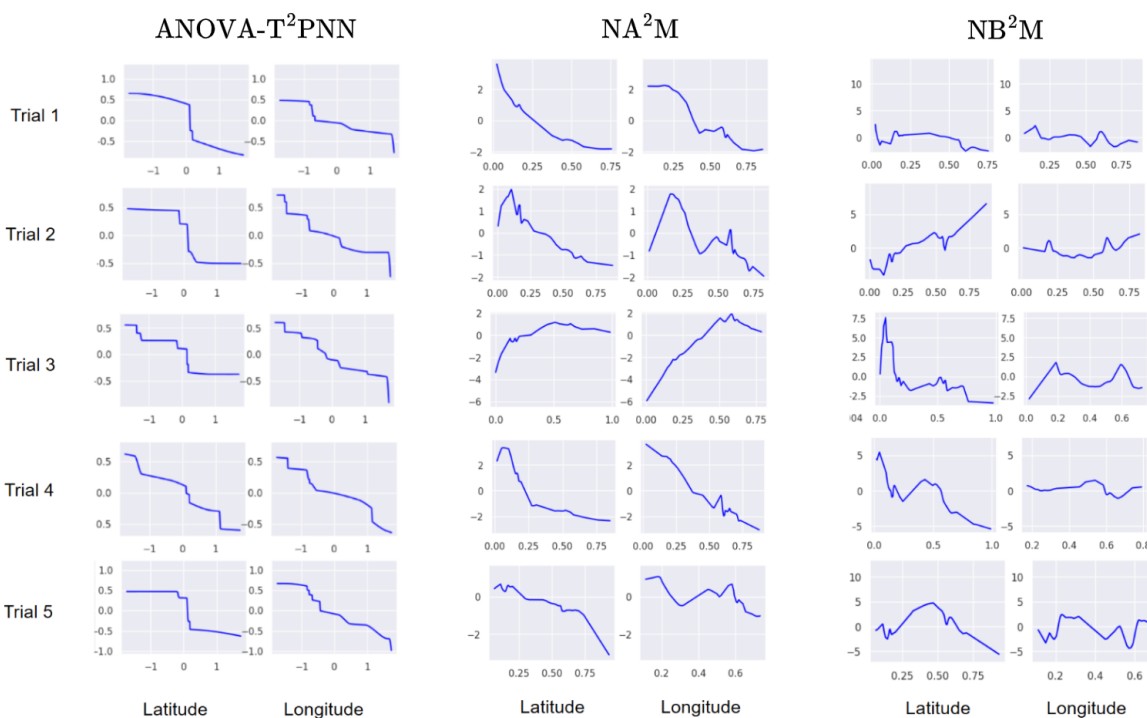

Figure 21: **Plots of the functional relations of 'Latitude' and 'Longitude' of CALHOUSING dataset after post-processing.**

We have seen that NA$^2$M and NB$^2$M are competitive to ANOVA-TPNN in prediction performance even though they are poor in estimating the components. There is a way to transform any estimates of the components to those that satisfy the sum-to-zero condition (Lengerich et al., 2020).

We consider an arbitrary estimated GA$^d$M $\hat{f}(\mathbf{x}) = \beta_0 + \sum_{S \subseteq [p], |S| \leq d} \hat{f}_S(\mathbf{x}_S)$. We fix $\mathbf{x}$ and write $f_S$ instead of $f_S(\mathbf{x}_S)$ for notational simplicity.

For given $S$ with $|S| = d$, we first transform $\hat{f}_S$ into

$$\tilde{f}_S = \hat{f}_S + \sum_{k=1}^{d} \sum_{V \subseteq S, |V|=k} (-1)^{d-k} \int_{\mathcal{X}_V} \hat{f}_S d\Pi_{j \in V} \mu_j. \tag{12}$$

Then, $\tilde{f}_S$ satisfies the sum-to-zero condition (Hooker, 2007; Lengerich et al., 2020). In turn for any $S' \subset S$ we redefine $\hat{f}_{S'}$ into

$$\hat{f}_{S'} = \hat{f}_{S'} - (-1)^{|S'|} \int_{\mathcal{X}_{S \setminus S'}} \hat{f}_S d\Pi_{j \in S \setminus S'} \mu_j. \tag{13}$$

We apply the transformation (12) to $\hat{f}_S$ for $|S| = d-1$ to have $\tilde{f}_S$, and redefine $\hat{f}_{S'}$ for $S' \subset S$ by (13). We repeat this process sequentially until $|S| = 1$ to have $\tilde{F}_S$ for all $S \subset [p]$ with $|S| \leq d$ which satisy the sum-to-zero condition.

Computational complexity of this post-processing for a fixed input is $\mathcal{O}(dn^{d-1})$ and thus computational complexity of calculating $\tilde{f}_S$ for all training data becomes $\mathcal{O}(dn^d)$ which is demanding when $n$ or $d$ is large. Furthermore, performing the post-processing requires storing thw whole dataset, which causes memory efficiency issues.

Table 26 compares the stability scores of the main effects of 'Latitude' and 'Longitude' of CALHOUSING dataset estimated by ANOVA-T$^2$PNN and post-processed NA$^2$M and NB$^2$M, and Figure 21 draws the 5 functional relations of the estimated main effects of 'Latitude' and 'Longitude' on 5 randomly sampled training data. It is observed that NA$^2$M and NB$^2$M are still unstable even after the post-processing, which suggests that instability in NAM and NBM is not only from unidentifiability but also instability of DNN.

## N. Details of Spline-GAM

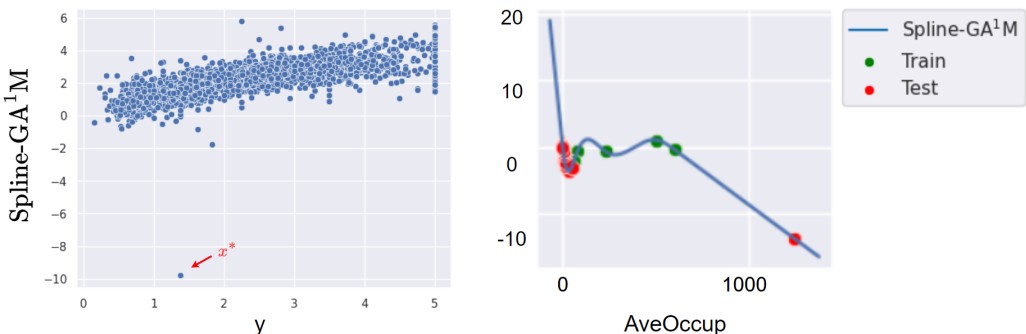

Figure 22: **Scatter plot of $(y_i^{test}, \hat{f}_{\mathrm{Spline\text{-}GA^1M}}(x_i^{test}))$, and the plot of the main effect 'AveOccup' estimated by Spline-GA$^1$M.**

Table 27: **Contribution of the main effects in Spline-GA$^1$M.**

| Main effect | 'MedInc' | 'HouseAge' | 'AveRooms' | 'AveBedrms' | 'Population' | 'AveOccup' | 'Latitude' | 'Longitude' |
|---|---|---|---|---|---|---|---|---|
| Contribution | 1.64 | 0.36 | -4.18 | 4.82 | 0.25 | -13.78 | -0.47 | 1.55 |

In this section, we describe Spline-GAM (Wood, 2017) which estimates each component by tensor product spline basis functions. For $j \in [p]$, let $t_{j,1}, ..., t_{j,M_j+5}$ be sorted knots into non-decreasing order on $\mathcal{X}_j$. For a given sequence of knots, the cubic spline basis functions for $f_j(\cdot)$ are defined as

$$B_{j,i,0}(x) = \mathbb{I}(t_i \leq x < t_{i+1})$$
$$B_{j,i,k}(x) = \frac{x - t_i}{t_{i+k} - t_i} B_{j,i,k-1}(x) + \frac{t_{i+k+1} - x}{t_{i+k+1} - t_{i+1}} B_{j,i+1,k-1}(x)$$

for $k = 1, 2, 3$ and $i = 1, ..., M_j$.

For general $S \subseteq [p]$, Spline-GAM estimates $f_S(\mathbf{x}_S)$ by

$$f_S(\mathbf{x}_S) \approx (\otimes_{j \in S} \mathbf{B}_j(x_j)) \boldsymbol{\alpha}_S$$
$$= g_S(\mathbf{x}_S)$$

where $\mathbf{B}_j(x_j) = (B_{j,1,3}(x_j), ..., B_{j,M_j,3}(x_j))^\top$ and the second derivatives of the basis functions at the boundaries are set to be zero. These basis functions are called the Natural Cubic Splines (Hastie, 2009).

Furthermore, to estimate each component in the functional ANOVA model by the smooth function, Spline-GAM employs the penalty term. For $S \subseteq [p]$, the penalty term for $f_S(\mathbf{x}_S)$ is defined as

$$J(g_S) = \lambda_S \sum_{j \in S} \int \left( \frac{\partial^2 g_S(\mathbf{x}_S)}{\partial^2 x_j} \right)^2 d\mathbf{x}_S$$

The smoothness of the spline model can be adjusted by tuning $\lambda_S > 0$.

**Inaccurate prediction beyond the boundary.** Spline-GAM is a model that sets knots based on the training data and interpolates between the knots using cubic spline basis functions. If the test data contains input outliers, the prediction performance of Spline-GAM may deteriorate for those outliers.

We conduct additional analysis on CALHOUSING dataset to examine the effect of input outliers. Figure 22 shows the scatter plot of $(y_i^{test}, \hat{f}_{\mathrm{Spline\text{-}GA^1M}}(x_i^{test}))$, $i = 1, ...., n_{\mathrm{test}}$, and the plot of the functional relation of the main effect 'AveOccup' estimated by Spline-GA$^1$M. In scatter plot, we observe that for a given data point $x^* = (10.2, 45, 3.17, 0.83, 7460, 1243, 38.32, -122)^\top$, the value of $y^*$ is approximately 1.3, while the corresponding prediction from Spline-GA$^1$M is approximately -10.

To investigate the reason behind such an inaccurate prediction for the data point $x^*$, we explore the contribution of each main effect in Spline-GA$^1$M as shown in Table 27. We observe that the contribution of the main effect 'AveOccup' is abnormally high. Upon examining the plot of the main effect 'AveOccup' estimated by the Spline-GA$^1$M, we conclude that the inaccurate prediction arises because the 'AveOccup' feature value of $x^*$ is an outlier and linear extrapolation of the cubic spline basis functions outside the range of inputs is used. In contrast, the TPNN with the sigmoid activation is bounded outside the range of inputs and so robust to input outliers.

