# OpenReview forum: "Tensor Product Neural Networks for Functional ANOVA Model"
_ICML.cc/2025/Conference — ICML 2025 poster_

### Official Review · Reviewer_9V8Z · 2025-03-12

**Overall Recommendation:** 4

**Summary:**

The authors propose an approach for learning functional ANOVA decompositions form data. The neural network-based architecture they propose is designed such that it admits a unique functional ANOVA decomposition. The authors prove their architecture is a universal approximator for smooth functions which satisfy the sum-to-zero condition. By approximating a unique functional ANOVA decomposition (rather than a non-unique functional ANOVA decomposition) the authors seek to make the learning process more stable. They show their approach provides comparable performance to standard approaches for XAI on some standard benchmarks while being more stable.

**Claims And Evidence:**

- A central motivating claim of their approach is that it is more stable than other methods for learning functional ANOVA decompositions from data. This claim is well supported by Tables 1, 2, and Appendix C.2.
- The authors claim that their approach can approximate a class of smooth functions well. This is supported by their theoretical result showing universal approximation as well as the numerical studies where their approach performs similarly to other SOTA methods for XAI.
- One of the claims which motivates the need for the sum-to-zero condition is that without identifiability, components become unstable and inaccurate. The study provided in Appendix F.1 was not convincing in my opinion. In particular, it wasn't clear to me what was meant by unstable and inaccurate in this case.

**Essential References Not Discussed:**

NA

**Experimental Designs Or Analyses:**

- Study 4.1 shows that the author's approach is preferred in terms of component estimation stability
- Study 4.2 shows that the proposed approach tends to learn a representation that is close to the true functional ANOVA on synthetic data. While this study is convincing, it would have been helpful to have included a study on a synthetic benchmark which does not satisfy the sum-to-zero condition by construction to understand how your approach might perform in less favorable situations.
- Study 4.3 shows that the proposed approach achieves comparable or better predictive performance to methods from the literature across a number of standard benchmark problems. The fact the proposed approach is no longer doing far better than standard approaches (like in Study 4.2) calls into question how realistic the assumption of the sum-to-zero condition is in practice. Some discussion on this would have been helpful.
- Study 4.4 applies the proposed approach to some high-dimensional problems convincingly demonstrating the approach can be useful on moderately sized problems.
- Study 4.5 compares the proposed approach to Spline-GAM (Serven 2018). This is an interesting study which demonstrates that the proposed approach seems to be more robust to outliers than this prior approach.
- Study 4.6 compares ANOVA-TPNN to NBM-TPNN. I think this study was comparatively weak. It would be useful to understand the computation time advantages of NBM-TPNN vs ANOVA-TPNN vs some standard approach for learning functional ANOVA decompositions.

**Methods And Evaluation Criteria:**

The authors evaluate their proposed approach on a number of synthetic benchmarks (consisting of three test functions) as well as a 13 real-world datasets. These datasets encompass a breadth of classification and regression problems making them well-suited for evaluating the proposed approach in my perspective.

**Other Comments Or Suggestions:**

NA

**Other Strengths And Weaknesses:**

- The proposed approach is a novel contribution as far as I'm aware. As discussed previously, the numerical studies do a good job of supporting the author's main claims.
- It would have been helpful to include a more complete discussion of why learning such decompositions is useful for interpretability in the main text.

**Questions For Authors:**

- One claim you make is that a model that provides more stable estimates of component functions is more desired in XAI. It was unclear to me why you might prefer one particular ANOVA decomposition over another if their predictive accuracies are similar. In other words, can you provide some additional insight into why the particular sum-to-zero condition is desirable for XAI over the many other methods for enforcing some form of identifiability (i.e. through regularization, etc.)?
- As a reader less familiar with XAI, can you provide a practical example of how one might use a learned ANOVA-TPNN model to drive decision making?

**Relation To Broader Scientific Literature:**

This work relates broadly to work on learning interpretable representations from data.

**Theoretical Claims:**

The sketch of the proof for Theorem 3.3 looks correct but I found the details in Appendix A to be a bit challenging to follow. It would be helpful if you provided references to some standard results that you rely on (even if they are textbook results).

---

> ### Author Rebuttal · Authors · 2025-04-01
>
> Thank you for your valuable and insightful feedback.
> We have made every effort to address your comments.
> Due to character limits, "Comment" is abbreviated as "C".
>
> >**C1 in Claims and Evidence** : One of the claims which...
>
> >**C2 in Supplementary Material :** I struggled to understand...
>
> **Response to C1 and C2.**
>
> We do not claim that component estimation is accurate under the sum-to-zero condition, but rather that it is stable.
> The sum-to-zero condition is not a unique condition to ensure the identifiability of each component in the functional ANOVA model.
> In turn, different conditions result in different component estimations, and thus considering the accuracy of estimating component would not make sense.
>
> However, stability is crucial since we want that the interpretation to be robust to data perturbation.
> Appendix F aims to visually demonstrate the superior stability of ANOVA-TPNN over NAM and NBM, as measured by the stability score in Section 4.1.
> Figures 5–16 show that the main effects estimated by ANOVA-TPNN are consistent across all trials, while those from NAM and NBM vary significantly.
>
> The main effects in the functional ANOVA model are useful for visual interpretation.
> Post-hoc methods (e.g., PDPs, SHAP) generate interpretation after model fitting.
> In contrast, ANOVA-TPNN is an in-processing method that jointly performs model estimation and interpretation, ensuring consistency between the model and interpretation plots.
>
> >**C3 in Theoretical Claims.** The sketch of the proof for...
>
> **Response.**
>
> The basis neural network in Equation (3) is inspired by a smooth version of a decision tree, where the indicator function is replaced with sigmoid functions.
> Thus, the sum of TPNNs resembles the sum of smooth decision trees. We used techniques in Lemma 3.2 of [1] to derive the approximation property of TPNN.
>
> >**C4 in Experimental Designs Or Analyses.** Study 4.2 shows...
>
> **Response.**
>
> The sum-to-zero condition is not a requirement for the true function.
> Rather, any functional ANOVA decomposition can be redecomposed into one that satisfies the sum-to-zero condition (See Section 22 in [2]).
>
> >**C5 in Experimental Designs Or Analyses.** Study 4.3 shows...
>
> **Response.**
>
> The smaller performance difference in Section 4.3 compared to Section 4.2 is not due to the sum-to-zero condition in the synthetic dataset, but rather due to the different evaluation criteria: Section 4.2 focuses on component selection, while Section 4.3 evaluates prediction performance.
>
> NA$^{2}$M and NB$^{2}$M perform poorly in component selection because they fail to properly separate main effects and second-order interactions.
> As shown in Figures 9 and 10, when second-order interactions are included, main effects in NA$^{2}$M and NB$^{2}$M are absorbed into the interactions, resulting in near-constant main effects.
> In contrast, ANOVA-T$^{2}$PNN uses the sum-to-zero condition, ensuring mutual orthogonality of components in the $L_2$ space, leading to more accurate identification of component effects, as shown in Figure 8.
>
> >**C6 in Experimental Designs Or Analyses.**
> Study 4.6 compares ...
>
> **Response.**
>
> See response to Reviewer 6BRT's W1.
>
> >**C6 in Supplementary Material.**
> Appendix D was very...
>
> >**C7 in Other Strengths And Weaknesses**
> It would have been...
>
> **Response to C6 and C7.**
>
> In response to the reviewer’s comments, we will move some contents from Appendix D to the main text and provide a more detailed explanation of the interpretability of the functional ANOVA decomposition in the final version of the manuscript.
>
> >**C8 in Questions For Authors.**
> One claim you...
>
> **Response.**
>
> As the reviewer mentioned, other identifiability conditions exist.
> Among them, the sum-to-zero condition is adopted for two main reasons.
>
> First, the sum-to-zero condition is easy to implement during training.
> For example, consider a functional ANOVA model for main effects:
> $ f(\mathbf{x}) = \sum_{j=1}^p f_j(x_j),$
> where $\mathbf{x} = (x_1,..,x_p)^\top$.
> We may consider the identifiability condition as $\forall i \neq j$, $\mathbb{E}[f_i(X_i)f_j(X_j)]=0$, but enforcing this for neural networks is difficult and the optimization is impractical.
>
> Finally, since ANOVA-TPNN satisfies the sum-to-zero condition, it enables fast and efficient computation of SHAP values using Proposition 3.2.
>
> >**C9 in Questions For Authors.**
> As a reader less...
>
> **Response.**
>
> As shown in Appendix E, by replacing the classifier in CBM with ANOVA-TPNN, the image model can be interpreted through the components estimated by ANOVA-TPNN.
> For a given image, the contributions of concepts can be determined as in Table 20, and importance scores can be calculated as in Tables 14 and 15 to identify which concepts the model considers important for classification.
>
> **References**
>
> [1]. Ročková et al. Posterior concentration for Bayesian regression trees and forests.
>
> [2]. Christoph, Molnar. Interpretable machine learning: A guide for making black box models explainable.

---

### Official Review · Reviewer_CFrs · 2025-03-13

**Overall Recommendation:** 3

**Summary:**

The paper proposes an approach for constructing interpretable machine learning models based on the functional ANOVA decomposition. The authors consider a decomposition of small order (1-2), and the decomposition terms are constructed with the basis functions represented by neural networks. To satisfy the condition of uniqueness of the expansion terms, the authors impose a natural restriction that the integral of each of the decomposition terms is equal to zero. The last condition is achieved by a special choice of the coefficient in the basis function.

**Claims And Evidence:**

In the introduction to your paper, you explicitly formulate the problem of interpretability of AI models. In this context, the task seems to consist of analyzing already existing and trained large neural network models. It is not immediately clear from the introduction that instead you are developing a directly interpretable model. In this context, the relevant questions raised in "Questions For Authors".

**Essential References Not Discussed:**

References to relevant works are provided, but perhaps a more detailed discussion of the innovations proposed in comparison with older works is missing. Also in the context of ANOVA decomposition it is probably logical to cite the well-known work of Sobol (2001).

**Experimental Designs Or Analyses:**

The authors conduct a comparison with modern alternative approaches on standard test datasets.

**Methods And Evaluation Criteria:**

Yes

**Other Comments Or Suggestions:**

--

**Other Strengths And Weaknesses:**

--

**Questions For Authors:**

1. I would ask you to formulate more clearly what exactly you consider to be the main innovation proposed in your approach (in comparison with previous works).
2. Can this approach be used to interpret already trained neural network models (for example, as neural network attribution methods)?
3. If the model you build is interpretable, then the question arises about demonstrating that interpretability and its usefulness. I did not see any examples of this in the experiments section.

**Relation To Broader Scientific Literature:**

The authors develop the approach proposed in works "Neural Basis Models for Interpretability" (2022) and Scalable Higher-Order Tensor Product Spline Models (2024).

**Theoretical Claims:**

Yes

---

> ### Author Rebuttal · Authors · 2025-04-01
>
> Thank you for your valuable feedback and questions.
> We have made every effort to address your insightful questions.
>
> > **Weakness 1 in Claims And Evidence.** In the introduction to your paper, you explicitly formulate the problem of interpretability of AI models. In this context, the task seems to consist of analyzing already existing and trained large neural network models. It is not immediately clear from the introduction that instead you are developing a directly interpretable model.
> In this context, the relevant questions raised in "Questions For Authors".
>
> **Response to Weakness 1.**
>
> In Line 12 on the right column of Page 1 of the manuscript, we mentioned that the functional ANOVA model is a transparent box-design model frequently used in explainable AI (XAI).
> Then, in Line 53 on the right column of Page 1, we stated that we propose a learning algorithm that estimates the components of the Functional ANOVA model using Tensor Product Neural Network (TPNN).
>
> In response to the reviewer’s comments, we will revise the introduction to explicitly state that "we propose a new transparent box-design model based on the functional ANOVA model and a specially designed neural network called Tensor Product Neural Network (TPNN)''  to improve clarity.
>
> $\newline$
>
> > **Weakness 2 in Essential References Not Discussed.**
> References to relevant works are provided, but perhaps a more detailed discussion of the innovations proposed in comparison with older works is missing. Also in the context of ANOVA decomposition it is probably logical to cite the well-known work of Sobol (2001).
>
> **Response to Weakness 2.**
>
> In Section 4.5, we compared ANOVA-TPNN with Spline GAM (older work), and found that our model is more robust to input outliers. For more details, please refer to Section 4.5 and Appendix O of the paper.
> As suggested by the reviewer, we will include references to key works on functional ANOVA decomposition, such as Sobol (2001).
>
>
> $\newline$
>
> >**Q1.** I would ask you to formulate more clearly what exactly you consider to be the main innovation proposed in your approach (in comparison with previous works).
>
> **Response to Q1.**
>
> The main innovation is a specially designed neural network called **Tensor Product Neural Network** (TPNN) defined in Equation (4) in Page 4.
> This neural network is not only flexible enough to satisfy the universal approximation property, as proven in Section 3.3, but also automatically satisfies the sum-to-zero condition without imposing any constraints on the learnable parameters, which allows the use of standard gradient descent algorithms.
> The sum-to-zero condition theoretically guarantees the uniqueness of the functional ANOVA decomposition (Proposition 3.1 of the paper), and thus it is essential for the stable estimation of each component, as demonstrated in Section 4.1.
> Existing deep neural network-based functional ANOVA models, such as NAM and NBM, do not satisfy the sum-to-zero condition, and thus they are unstable in estimating components.
> **ANOVA-TPNN is the first neural network which is flexible (e.g. having the universal approximation property) but satisfies the sum-to-zero condition without any additional constraints.**
>
> Next, unlike traditional tensor product basis expansion approaches, TPNN does not lead to an exponential increase in the number of learnable parameters when estimating component functions $f_{S}$ as $|S|$ increases.
> For more details, please refer to the **Remark** on Page 4 of the paper.
>
> Furthermore, since ANOVA-TPNN satisfies the sum-to-zero condition, it allows for fast and accurate computation of SHAP values by leveraging Proposition 3.2 of the paper.
>
> $\newline$
>
> >**Q2.** Can this approach be used to interpret already trained neural network models (for example, as neural network attribution methods)?
>
> **Response to Q2.**
>
> Yes, it is possible.
> A pre-trained neural network can be approximated using ANOVA-TPNN by treating the prediction values of a pre-trained neural network as outputs of the training data, and interpretation can then be provided by analyzing the estimated components as is done in Appendix D of the paper.
> We will add post-hoc interpretation results of a pre-trained neural network using ANOVA-TPNN to Appendix D.
>
> $\newline$
>
> >**Q3.** If the model you build is interpretable, then the question arises about demonstrating that interpretability and its usefulness. I did not see any examples of this in the experiments section.
>
> **Response to Q3.**
>
> In Section 4.7 of the paper, we applied ANOVA-TPNN to image classification and illustrated how the prediction model can be interpreted based on the components estimated by ANOVA-TPNN.
> Due to the page limitations, details related to interpretations using ANOVA-TPNN are provided in Appendix D-E of the manuscript.
> For example, in Appendix E, we described the results of both local and global interpretations of the image classification model using the components estimated by ANOVA-TPNN.

---

> > ### Comment · Reviewer_CFrs · 2025-04-03
> >
> > I thank the authors for their detailed comments. Regarding your response to Q2, I think the results of such post-hoc interpretation experiments would be really interesting to add to the appendices. Your response dispelled my doubts, and I believe that I should increase the rating of your work (2 -> 3).

---

> > > ### Author Response · Authors · 2025-04-04
> > >
> > > Thank you for your thoughtful response and for reconsidering your rating of our work.
> > > We appreciate your suggestion regarding the post-hoc interpretation experiments, and we will incorporate these results into the appendix to further strengthen our paper.
> > > Once again, we sincerely appreciate your time and constructive feedback.

---

### Official Review · Reviewer_6BRT · 2025-03-14

**Overall Recommendation:** 3

**Summary:**

This paper introduces ANOVA Tensor Product Neural Network (ANOVA-TPNN), a novel neural network framework designed to estimate the functional ANOVA model with greater stability and accuracy. Theoretical analysis confirms that ANOVA-TPNN has universal approximation capabilities for smooth functions. Empirical studies across multiple benchmark datasets demonstrate that ANOVA-TPNN provides more stable component estimation and interpretation than existing models like NAM, NBM, NODE-GAM, and XGB. Additionally, the paper introduces NBM-TPNN, a variant that enhances scalability by ensuring the number of basis functions is independent of input feature dimensionality. Despite these advantages, the authors acknowledge computational challenges when handling high-order interactions, suggesting future work on component selection techniques.

**Claims And Evidence:**

All the claims made in the abstract are supported by clear and convincing evident.

**Essential References Not Discussed:**

Relevant works are cited through the papers.

**Experimental Designs Or Analyses:**

The experimental designs are soundness.

**Methods And Evaluation Criteria:**

The proposed method makes sense for the problem.

**Other Comments Or Suggestions:**

No.

**Other Strengths And Weaknesses:**

**Strengths:**

- The paper is clearly presented, and both theoretical and experimental justifications are provided.

- The paper addresses a crucial issue in explainable AI (XAI) by improving the stability of functional ANOVA decomposition.

- The authors provide a universal approximation proof, ensuring the validity of the proposed method.

- The model demonstrates competitive prediction accuracy while offering superior component stability compared to baseline models.

**Weaknesses:**

While ANOVA-TPNN improves efficiency over traditional basis expansion approaches, the paper acknowledges that high-order interactions remain computationally demanding. Additional analysis of runtime complexity would strengthen the work. Beside of this concern, I do not see any other major weaknesses of the paper.

**Questions For Authors:**

- Can NBM-TPNN be further extended to handle higher-order interactions efficiently, perhaps through sparsity-inducing techniques?

- How does the choice of activation functions and network architecture impact the stability of the estimated components?

**Relation To Broader Scientific Literature:**

This paper introduce ANOVA-TPNN, a novel neural network framework designed to estimate the functional ANOVA model in XAI.

**Theoretical Claims:**

I have not checked all of the proofs in detail. However, it seems to me that the proof for the universality is correct.

---

> ### Author Rebuttal · Authors · 2025-04-01
>
> Thank you for your valuable feedback and questions.
> We have made every effort to address your insightful questions.
> > **W1.** While ANOVA-TPNN improves efficiency ...
>
> **Response to W1.**
>
> We have conducted runtime experiments for the functional ANOVA model only with the main effects in Appendix K, whose results are summarized in Table 23.
> These results suggest that ANOVA-TPNN is competitive with other baselines in terms of runtime complexity.
>
> As the reviewer pointed out, we conducted additional experiments for the runtime complexity of higher order functional ANOVA models.
> We analyzed Abalone data with the functional ANOVA models with up to the 4th order interactions and compared the runtimes of ANOVA-TPNN, NAM and NBM.
> The hyperparameters of all models are set identically to those in Appendix K of the paper.
>
> **Table A.1**
> |Maximum order of interaction|1|2|3|4|
> |-|-|-|-|-|
> |NAM|6.6sec|11.1sec|28.3sec|79.3sec|
> |NBM | 3.0 sec|6.8sec|12.2sec|21.1sec|
> |ANOVA-TPNN|1.6sec|5.2 sec|22.7sec|82.7sec|
> |NBM-TPNN|1.5sec|4.1sec|7.8sec|16.4sec|
>
> Table A.1 presents the results, which amply show that ANOVA-TPNN is competitve with NAM and NBM in terms of runtime.
> In addition, it is interesting to see that the runtimes of NAM and ANOVA-TPNN is super-linear in the order of interactions while those of NBM and NBM-TPNN are linear.
>
> We emphasize that estimating high-order interactions is a common challenge in all functional ANOVA models, including NAM and NBM. To address this, we used Neural Interaction Detection (NID) to remove unnecessary components before training. Section 4.4 demonstrates its effectiveness through numerical experiments.
>
> > **Q1.** Can NBM-TPNN be further extended...
>
> **Response to Q1.**
>
> One may consider the group lasso penalty, which is well-suited for selecting meaningful components while promoting sparsity.
> NBM-TPNN models component $f_{S}$ as
>
> $f_S(\textbf{x}\_{S} ) =  \sum_{k=1}^{K} \beta_{S,k} \prod_{j \in S} \phi(x_{j} | \theta_{k}),$
>
> where $\beta_{S,k},\theta_{k}$ are learnable parameters and $\phi(\cdot)$ is the basis neural network which is defined in Section 3.4 of the paper. Note that the parameters $\theta_k$ are shared by the components while
>  $\beta_{S,k}$ are not, which makes it possible to apply a sparse penalty.
>
> Given observed data ${ (y_{i},\textbf{x}\_{i}) }\_{i=1}^{n}$ with $\mathbf{x}\_{i}=(x_{1,i},...,x_{p,i})^\top \in \mathbb{R}^p$ and $y_i \in \mathbb{R}$, consider the objective:
>
>  $ {1\over n}\sum_{i=1}^n\bigg (y_{i} - \sum_{S \subseteq [p],|S|\leq d}f_S(\mathbf{x}\_{S,i}) \bigg )^2 +  \sum_{S \subseteq [p],|S|\leq d}\lambda_{S}\Vert \mathcal{B}\_{S}\Vert_2,$
>
> where $\lambda_{S} > 0$ is a hyperparameter, $\mathcal{B}\_{S} = (\beta\_{S,1},...,\beta\_{S,K})^\top$ , $\mathbf{x}\_{S,i} = (x_{j,i}, j \in S)$ and $d$ is the highest order of interactions.
> Then, the group lasso penalty makes $\mathcal{B}\_{S}$ sparse on the component level, enabling component selection during training. This sparse estimation improves interpretability of higher-order interactions.
>
> However, group lasso penalty does not help reduce runtime complexity. The number of parameters remains proportional to $Kp^{d}$, regardless of sparsity. Developing new algorithms using forward selection or random search could be a promising future direction for future work.
>
> > **Q2.** How does the choice...
>
> **Response to Q2.**
>
> The experimental results on the prediction performance and stability of ANOVA-TPNN when using the ReLU activation function are already given in Appendix L of the paper.
> We found that ReLU slightly underperforms compared to the sigmoid version in prediction accuracy and component stability, likely because the sigmoid-based TPNN is more robust to input outliers.
>
> As the reviewer suggested, it is interesting to investigate how the choice of a different network architecture, rather than our proposed TPNN model, affects the stability of component estimation.
> Therefore, we conducted additional experiments to evaluate the choice of the network architecture on the performance of stability.
> We consider a deep neural network based tensor product model (TPDNN) which assumes
>
> $ f_{S}(\textbf{x}\_{S}) = \prod_{j \in S}g(x_{j} | \theta_{j,S})$
>
> for each component $f_S,$ where $g(\cdot|\theta_{j,S}) : \mathbb{R} \xrightarrow{} \mathbb{R}$ is a 3-layer neural network with hidden sizes [32,32,16] and $\theta_{j,S}$s are the learnable parameters.
> We refer to the model that estimates components up to order $d$ in the functional ANOVA model using a TPDNN as ANOVA-T$^{d}$PDNN.
>
> **Table A.2**
> |Model|ANOVA-T$^{2}$PNN|ANOVA-T$^{2}$PDNN|
> |-|-|-|
> |RMSE|2.087(0.08)|2.148(0.08)|
> |Stability score|0.028|0.041|
>
> As in Section 4.3, we ran 10 trials.
> Table A.2 shows the averaged prediction and stability scores for ANOVA-T$^{2}$PNN and ANOVA-T$^{2}$PDNN on the Abalone dataset.
> ANOVA-T$^{2}$PNN outperforms ANOVA-T$^{2}$PDNN, likely due to TPNN's robustness to input outliers.
> We will add these results to the Appendix.

---

### Decision · Program_Chairs · 2025-05-01

**Decision:**

Accept (poster)

**Comment:**

The paper proposes an approach for constructing interpretable machine learning models based on the functional ANOVA decomposition. The paper is clearly presented, and both theoretical and experimental justifications are provided. Some concerns on the runtime analysis and clarity of presentation have been successfully addressed by the authors' rebuttal, which should be incorporated into the camera-ready version.